# Redefining Chemotherapy-Related Headaches: From Pathobiology to Differential Diagnosis and Management

**DOI:** 10.3390/ijms27010262

**Published:** 2025-12-26

**Authors:** Chioma V. Nnadi, David B. Olawade, Susan Shorter, Emmanuel O. Oisakede, Stergios Boussios, Saak V. Ovsepian

**Affiliations:** 1Faculty of Engineering and Science, University of Greenwich, London ME4 4TB, UK; 2Department of Allied and Public Health, School of Health, Sport and Bioscience, University of East London, London E16 2RD, UK; d.olawade@uel.ac.uk; 3Department of Research and Innovation, Medway NHS Foundation Trust, Gillingham ME7 5NY, UK; 4Department of Business, Management and Health, York St John University, London E14 2BA, UK; 5School of Health and Care Management, Arden University, Arden House, Middlemarch Park, Coventry CV3 4FJ, UK; 6Department of Clinical Oncology, Leeds Teaching Hospitals Trust, Leeds LS9 7TF, UK; 7Department of Health Research, University of Leeds, Leeds LS2 9JT, UK; 8Department of Medical Oncology, Medway NHS Foundation Trust, Gillingham ME7 5NY, UK; stergios.boussios@nhs.net; 9Faculty of Medicine, Health, and Social Care, Canterbury Christ Church University, Canterbury CT2 7PB, UK; 10Faculty of Life Sciences & Medicine, School of Cancer & Pharmaceutical Sciences, King’s College London, Strand, London WC2R 2LS, UK; 11Kent Medway Medical School, University of Kent, Canterbury CT2 7LX, UK; 12AELIA Organisation, 9th Km Thessaloniki-Thermi, 57001 Thessaloniki, Greece; 13Faculty of Medicine, School of Health Sciences, University of Ioannina, 45110 Ioannina, Greece; 14Department of Medical Oncology, University Hospital of Ioannina, 45500 Ioannina, Greece; 15Faculty of Medicine, Tbilisi State University, Tbilisi 0179, Georgia

**Keywords:** chemotherapy-related headaches, neuroinflammation, blood-brain barrier disruption, systemic toxicity, pain management

## Abstract

Chemotherapy-related headaches pose a significant challenge to the well-being and treatment adherence of cancer patients. Despite their prevalence, the underpinning mechanisms and pathobiology remain elusive, limiting treatment options. Herein, we review emerging causes, molecular and functional processes, and mechanisms at play, and discuss research and clinical gaps. We consider the iatrogenic and psychogenic effects of chemotherapy and highlight the need to distinguish chemotherapy-related headaches from primary headache disorders in cancer patients, including migraines or tension-type headaches. We discuss evolving biomarkers and mechanistic models that could facilitate the differential diagnosis and development of effective interventions. Given the global rise of cancer burden and better outcomes of chemotherapy with longer life expectancy, recognition of the detrimental impact of chemotherapy-related headaches and their integration into management plans are expected to improve treatment adherence and post-treatment life quality.

## 1. Introduction

Chemotherapy encompasses a diverse range of pharmacological treatments designed to selectively target and eliminate rapidly dividing cancer cells through various mechanisms, including DNA alkylation, topoisomerase inhibition, antimetabolite activity, and microtubule disruption. While this approach has significantly improved survival rates and disease management, it is not without complications. The limited selectivity of chemotherapy results in harmful effects that extend beyond cancerous cells, influencing the functions of various organs and systems of the recipient. The nervous system is particularly vulnerable, with off-target effects leading to a spectrum of neurological complications that can impair sensory, motor and cognitive functions [1,2,3]. These outcomes pose significant challenges for treatment adherence and maintaining patients’ quality of life and are the subject of in-depth research [4,5].

Among the neurological complications, chemotherapy-related headaches (CRHs) are one of the most prevalent yet least addressed. This review represents the first comprehensive effort to establish CRHs as a distinct clinical entity requiring specific diagnostic and therapeutic approaches, rather than treating them as merely incidental symptoms of broader neurotoxic effects. Headaches can significantly impact cancer patients, leading to increased discomfort, reduced treatment compliance, and contributing towards deterioration of patient well-being [1,4]. The prevalence of CRHs varies depending on several factors, including the type and dosage of administered chemotherapy, treatment duration, and individual patient susceptibility. Table 1 summarizes the reported incidence of CRHs relative to other common neurological complications, demonstrating that headaches occur with a frequency comparable to, or greater than, that of many widely recognized adverse effects. Notably, subjects with a history of headaches are at an increased risk of developing more severe or persistent CRHs [4,6]. Despite prevalence, CRHs remain underrecognized and overshadowed by other neurological adverse effects of chemotherapy such as stroke, neuropathy, myelopathy, seizure and cognitive deficit [3,7] (Figure 1). The lack of effective treatments for CRHs is contributed to by the scarcity of mechanistic and translational studies exploring their cause, progression, and implications for the prognosis of cancer.

Unlike primary headache disorders such as migraine and tension-type headaches, which have well-established diagnostic criteria outlined in the International Classification of Headache Disorders, Third Edition (ICHD-3) [23], CRHs lack specific classification parameters. Primary headaches are characterized by distinct pathophysiological mechanisms independent of underlying systemic disease. In contrast, CRHs represent a secondary headache disorder directly linked to chemotherapeutic neurotoxicity and its cascade effects on neural and vascular systems. Importantly, CRHs must also be differentiated from headaches caused by intracranial metastases, leptomeningeal disease, increased intracranial pressure, or paraneoplastic syndromes, which implicate different mechanisms and therapeutic strategies. Additionally, headaches following intrathecal chemotherapy administration, while sharing some features with CRHs, represent a distinct subcategory characterized by acute meningeal irritation [24].

The pathophysiology of CRHs remains elusive, but emerging evidence suggests a multifactorial origin involving both iatrogenic and psychogenic effects. Iatrogenic effects result primarily from chemotherapy-induced neurotoxicity, which leads to inflammatory response, immune suppression, neuronal and glial dysfunctions and alterations in cerebrospinal fluid (CSF) with secondary effects. These alterations contribute to brain tissue damage and heightened pain sensitivity, leading to an increased likelihood of headaches that may resemble migraines [4,25]. Additionally, chemotherapy-induced alterations in vascular function and neurotransmitter balance may exacerbate the onset and severity of headaches. On the other hand, psychogenic effects such as anxiety, depression, and sleep disturbances, which are common among cancer patients undergoing intensive treatment regimens, have been implicated in worsening CRHs. The interplay between an array of pharmacological and psychological factors can make CRHs a complex and multifaceted neurological syndrome that requires further mechanistic and translational studies [1,4,26].

Notwithstanding their detrimental effects on patient well-being and treatment adherence, there is a significant gap in understanding CRHs and a pressing need for their targeted treatments. Most patients with CRHs rely on headache management with general painkillers and nonsteroidal anti-inflammatory drugs (NSAIDs), which may provide symptomatic relief but fail to address the underlying mechanisms responsible for headache development [4,26]. In this review, we analyze the literature on the epidemiology, pathophysiology, and management of CRHs, and identify gaps warranting further investigation. Its novelty is its focus on CRHs as a distinct clinical entity rather than as a secondary or incidental symptom of the neurotoxic effects of chemotherapy. By systematically evaluating existing research, we present a comprehensive summary of the emerging mechanisms of CRHs, their presentation, and the challenges and opportunities in diagnosis and treatment. It emphasizes the importance of integrating CRH management into cancer care protocols where appropriate to enhance patient well-being and treatment adherence. Finally, we propose that CRHs be recognized as a distinct clinical entity in future revisions of the ICHD, accompanied by a clearly defined diagnostic and management protocol, even as ongoing research continues to elucidate its underlying pathophysiology.

## 2. Adverse Neurological Effects of Chemotherapy—A Brief Overview

This section provides essential context for understanding CRHs within the broader spectrum of chemotherapy-induced neurotoxicity. While subsequent sections detail the specific mechanisms unique to CRHs, here, we briefly outline the general landscape of neurological complications to establish the clinical significance of headaches relative to other adverse events. While essential for treating cancer, chemotherapy often comes with a significant trade-off in the form of adverse neurological effects due to drug-induced toxicity. Despite the protective role of the blood–brain barrier (BBB), various chemotherapeutic agents can exert neurotoxic effects, occasionally leading to a range of complications [1,3,27]. These arise from diverse pathophysiological mechanisms and manifest in multiple clinical forms, including cognitive impairment, seizure, stroke, myelopathy, peripheral neuropathy, and persistent mild to severe headaches [6,21,28,29,30] (Figure 1). While several of these neurological effects have been extensively reported, CRHs remain largely overlooked despite their prevalence and impact on the quality of life.

### 2.1. Cognitive Impairments

Cognitive dysfunction, commonly referred to as “chemo brain” or “chemo fog”, is a well-documented adverse effect of chemotherapy, affecting a significant proportion of cancer patients [12,13,14,15]. Symptoms include memory deficit, reduced attention span, impaired executive function, slowed information processing, and difficulties with verbal fluency and multitasking. These impairments can persist for months or even years after treatment completion, significantly impacting daily functioning and overall quality of life [13,31].

The underlying mechanisms of chemotherapy-induced cognitive impairments remain an area of intense research. Hypotheses include direct neurotoxicity, oxidative stress, neuroinflammation, disruption of hippocampal neurogenesis, and alterations in neurotransmitter systems [13,31]. Some agents, such as methotrexate, cisplatin, and 5-fluorouracil, have been shown to induce neuronal damage and synaptic dysfunction [32,33]. Additionally, chemotherapy can compromise the integrity of the BBB, allowing peripheral inflammatory mediators to enter the central nervous system (CNS) and exacerbate cognitive impairment [3,27]. Growing evidence suggests that cognitive rehabilitation, physical exercise, and pharmacological interventions, such as modafinil and methylphenidate, may help alleviate cognitive symptoms [34].

### 2.2. Seizures

Seizures are common and severe neurological complications of chemotherapy, particularly in patients receiving high-dose or intrathecal (IT) treatment [16,17,35]. Patients with a history of epilepsy or preexisting neurological conditions are at a higher risk of experiencing chemotherapy-induced seizures [18]. Seizures in cancer patients pose a significant challenge, as they can lead to treatment delays, increased hospitalizations, and a decline in overall prognosis. The incidence of chemotherapy-induced seizures varies depending on the specific agents used, with certain drugs known to have a higher epileptogenic potential. Chemotherapeutic agents associated with seizure activity include methotrexate, cytarabine, ifosfamide, and platinum-based compounds such as cisplatin [4,36]. The mechanisms through which these drugs induce seizures involve direct neuronal toxicity, disruption of ion channel function, mitochondrial deficiency, and excitotoxicity caused by excessive release of glutamate in the brain [37]. In some cases, chemotherapy-related metabolic disturbances, such as hypoglycemia, hyponatremia, or hypocalcemia, can contribute to seizure susceptibility [38] (Figure 1). Management strategies typically include antiepileptic drugs (AEDs) such as levetiracetam, valproate, or lamotrigine, although drug interactions with chemotherapy agents must be carefully considered to avoid cross-reaction or reduction in treatment efficacy [16,18].

### 2.3. Stroke

Chemotherapy has been linked to an increased risk of ischemic and hemorrhagic strokes, particularly in patients receiving angiogenesis inhibitors or platinum-based chemotherapeutic agents [19,20,28]. This risk is primarily associated with chemotherapy-induced endothelial dysfunction, prothrombotic states, vascular inflammation, and injurious effects on cerebral blood vessels [39] (Figure 1). Agents such as bevacizumab, cisplatin, and 5-fluorouracil have been implicated in chemotherapy-related strokes. Bevacizumab, a vascular endothelial growth factor (VEGF) inhibitor, has been shown to increase the risk of arterial thromboembolism, leading to ischemic stroke [40]. Conversely, cisplatin induces a hypercoagulable state by increasing platelet activation and impairing fibrinolysis, elevating the risk of adverse cerebrovascular events [41]. In cancer patients, strokes are particularly concerning because they often present atypical symptoms and are easily overlooked in the setting of chemotherapy-induced fatigue or cognitive dysfunction. Furthermore, the use of anticoagulants must be considered with caution due to the risk of chemotherapy-induced thrombocytopenia and bleeding complications [42]. Early detection through neuroimaging, aggressive management of risk factors (hypertension, diabetes, and hyperlipidemia), and the use of antiplatelet therapy in high-risk patients may help mitigate stroke risk in this population [43,44].

### 2.4. Myelopathy

Chemotherapy-induced myelopathy is a rare but devastating condition that affects the spinal cord, leading to progressive weakness, sensory loss, and autonomic dysfunction. Myelopathy can result from direct neurotoxicity, ischemic damage, or immune-mediated demyelination caused by chemotherapy agents [21,22,45] (Figure 1). Drugs such as methotrexate, cisplatin, and cytarabine have been implicated in the development of myelopathy [21]. Methotrexate, when administered intrathecally, has been associated with leukoencephalopathy and necrotizing myelopathy, conditions characterized by white matter degeneration and severe neurological disability [46]. Additionally, platinum-based drugs can induce spinal cord ischemia by disrupting the blood supply to the spinal vasculature [47]. Symptoms of chemotherapy-induced myelopathy often mimic other neurological disorders, leading to delays in diagnosis. Patients may present with gradual-onset lower limb weakness, numbness, loss of deep tendon reflexes, and, in severe cases, paralysis [48]. Autonomic dysfunction, including urinary retention and bowel incontinence, may also occur [49]. There is currently no established treatment for chemotherapy-induced myelopathy, and management is mainly palliative [50]. Early physical therapy, corticosteroid administration in immune-mediated cases, and symptom control with neuromodulator agents may help improve outcomes [51].

### 2.5. Neuropathy

Chemotherapy-induced peripheral neuropathy (CIPN) is one of the most common and debilitating neurological side effects of chemotherapy, affecting up to 70% of cancer patients [10,11]. CIPN presents as sensory, motor, and autonomic dysfunction, with symptoms including tingling, numbness, burning pain, muscle weakness, and loss of reflexes [52]. These symptoms can significantly impact a patient’s daily activities, mobility, and overall quality of life, often persisting long after chemotherapy.

Several chemotherapeutic agents have been strongly associated with the development of neuropathy. Platinum-based drugs such as cisplatin, oxaliplatin, and carboplatin primarily cause sensory neuropathy due to their toxic effects on the dorsal root ganglia, leading to persistent numbness and pain [53]. Taxanes, including paclitaxel and docetaxel, disrupt microtubule stability, impair axonal transport and nerve function, and result in sensory and motor deficits [54]. Vinca alkaloids, such as vincristine and vinblastine, interfere with mitotic spindle formation and axonal integrity, leading to neuropathic pain and weakness [55]. Additionally, bortezomib, a proteasome inhibitor, is known to cause painful small-fiber neuropathy, which can severely impact the comfort and function of patients [56]. The pathophysiology of CIPN is complex and involves multiple mechanisms (Figure 1). Mitochondrial dysfunction, oxidative stress, microtubule damage, and the release of inflammatory cytokines contribute to nerve injury and degeneration [55,57]. Unlike some chemotherapy-induced toxicities that resolve after treatment cessation, CIPN often persists for months or even years, leading to chronic pain and long-term disability [57,58]. This prolonged adverse effect can severely impact the post-treatment quality of life of cancer survivors, making it a significant concern in oncological care. Currently, treatment options for CIPN remain limited and largely symptomatic. Medications such as gabapentinoids (gabapentin, pregabalin), and serotonin-norepinephrine reuptake inhibitors (SNRIs), including duloxetine, and topical agents, like capsaicin, are commonly used to alleviate neuropathic pain [58]. However, these treatments do not address the underlying nerve damage, and their effectiveness varies among patients.

## 3. Chemotherapy-Related Headaches: A Review of Reported Cases

Associated with chemotherapy headaches range from mild discomfort to severe and persistent pain that complicates the cancer treatment process. The duration of headaches also varies, with some lasting only a few hours and others persisting for days, significantly impacting a patient’s daily function and overall well-being [26]. The prevalence of CRHs ranges widely, from ~10% to over 30%, depending on age, type of cancer, chemotherapy, and other variables [4,8].

CRHs can significantly disrupt a patient’s daily life, interfering with sleep, mood, and overall functionality [59,60]. Emerging evidence suggests a mechanistic overlap between migraine and CRHs, indicating shared pathways such as hormonal fluctuations, chronic inflammation, and transmitter dysregulation. Since inflammation is a key driver in both conditions, cancer patients with a preexisting history of migraines may experience more frequent and severe headaches during chemotherapy [61]. When compounded by other standard chemotherapy side effects, such as nausea, vomiting, and cognitive dysfunction, CRHs can become profoundly debilitating, posing additional challenges to the clinical management of cancer.

Systematic analysis of reported CRH cases reveals several shared mechanistic features that transcend specific cancer types. First, platinum-based compounds, particularly cisplatin and oxaliplatin, are consistently associated with vascular dysfunction and thrombotic events across multiple malignancies, which might contribute to headaches [62]. Second, methotrexate, especially when administered intrathecally, demonstrates a strong correlation with meningeal irritation, cerebrospinal fluid pressure changes [63] or resultant stroke regardless of the underlying cancer diagnosis [64]. Third, chemotherapy-induced stress can promote neuroinflammatory cascades characterized by elevated pro-inflammatory cytokines, such as TNF-α and IL-6 [65], which are implicated in chronic headaches. Fourth, patients with pre-existing migraine history demonstrate a higher risk of developing severe CRHs across all cancer types examined, with a significant impact on cancer outcome [9,61]. These unifying features suggest standard pathophysiological mechanisms that could be targeted therapeutically. As discussed below, CRHs prevail across a broad spectrum of malignancies, with their frequency and clinical presentation contributed by a complex interplay of factors, including cancer type, treatment regimen, patient age, and others.

### 3.1. Breast Cancer

A study by De Sanctis and co-workers reported that 29.8% of breast cancer patients experienced headaches ranging in intensity from mild to severe migraine-like episodes [66]. The cohort had a mean age of 53.5 years, with most subjects being hormone receptor-positive and undergoing various treatments, including chemotherapy (92%), endocrine therapy (66%), and radiotherapy (52%). In ~20% of subjects, the headaches worsened after systemic chemotherapy treatment. The impact of headaches on the patients’ well-being was substantial, as symptoms added to the overall treatment burden, negatively affecting quality of life and sometimes compelling treatment modifications (Table 2).

### 3.2. B-Cell Acute Lymphoblastic Leukemia (ALL)—Case 1

Kataria and co-workers described a 25-year-old male with B-cell ALL who developed severe neurological symptoms, including headache, vomiting, disorientation, and seizures, within 24 h of starting chemotherapy [67]. The symptoms were attributed to the neurotoxic effects of the MINE chemotherapy protocol (Mesna, Ifosfamide, Mitoxantrone, and Etoposide). The rapid neurological deterioration required immediate intervention, including the cessation of chemotherapy to prevent further complications. This case underscores the potential severity of neurotoxicity and associated headaches in young adults and the need for vigilant neurological assessment in patients receiving intensive regimens (Table 2).

### 3.3. B-Cell Acute Lymphoblastic Leukemia (ALL)—Case 2

Manoukian and co-workers documented a 2-year-old child with B-cell acute lymphoblastic leukemia (ALL) who presented with a frontal headache lasting one week, accompanied by visual disturbances [68]. Given the young age of the patient, headaches were difficult to assess objectively, and imaging was required to rule out CNS leukemia. The symptoms were suspected to be due to methotrexate-induced systemic neurotoxicity or possible CNS involvement of leukemia. This case study emphasizes the importance of differential diagnosis in pediatric leukemia patients, as early signs of CNS infiltration can mimic chemotherapy-induced side effects. The management of this young patient required a multi-disciplinary and complex approach involving neurologists and oncologists to balance cancer treatment with symptom control (Table 2).

### 3.4. Chronic Lymphocytic Leukemia (CLL)—Case 3

Abd-Elsayed and Pahapill reported a 56-year-old male with CLL who developed severe, persistent headaches following chemotherapy [69]. The etiology of the headache was suspected to be chemotherapy-induced neurotoxicity, possibly aggravated by the patient’s pre-existing conditions, such as hypertension. Due to the intensity and persistence of the symptoms, various pain management strategies, including corticosteroids and nerve blocks, were considered. This case highlights the challenge of managing neurotoxicity-related pain in leukemia patients, particularly those undergoing long-term treatment (Table 2).

### 3.5. Lymphoblastic Lymphoma—Case 4

A study by Hu and co-workers detailed a 10-year-old male diagnosed with lymphoblastic lymphoma who initially presented with numbness of the face and lips, which later progressed to systemic pain, including headaches [70]. These neurological symptoms were linked to vincristine-induced neurotoxicity, which is a well-known side effect of this chemotherapy agent. The inflammatory response to chemotherapy exacerbated the patient’s discomfort, requiring intervention with an analgesic pump for effective pain relief. The case illustrates the importance of early recognition and management of chemotherapy-induced neuropathic pain to improve patient comfort and adherence to treatment (Table 2).

### 3.6. Non-Hodgkin’s Lymphoma—Case 5

De La Riva and co-workers examined headache prevalence in non-Hodgkin lymphoma patients, reporting a headache frequency of 10.3% [71]. Headaches typically lasted between 4 and 36 h, with a median duration of 15 h. The primary mechanism was linked to IT chemotherapy involving methotrexate, cytarabine and hydrocortisone. These agents, used to prevent CNS relapse, are known to cause meningeal irritation, leading to persistent headaches. Given the recurrence and duration of headaches in these patients, the study highlighted the need for effective pain management strategies, including prophylactic analgesics, to improve treatment tolerability (Table 2).

### 3.7. Small-Cell Lung Cancer—Case 6

A case study led by Rai and co-workers described a 58-year-old male smoker diagnosed with small-cell lung cancer who experienced severe, holocranial headaches described as pressure-like, which were unresponsive to NSAIDs [6]. The symptoms were attributed to chemotherapy-induced neurotoxicity and intracranial pressure changes, possibly exacerbated by an underlying metastatic disease. Due to the lack of response to conventional pain relief, intravenous morphine was required. This case highlights the need for robust pain management approaches in lung cancer patients experiencing severe headaches, especially those with concurrent metastases or intracranial involvement (Table 2).

### 3.8. Testicular Cancer—Case 7

Clarke and co-workers reported a case of a 22-year-old male with testicular cancer who presented with moderate, persistent headaches that worsened with coughing and straining [72]. The headache was suspected to result from cisplatin-induced vascular effects, leading to cerebral venous sinus thrombosis (Table 2). This case highlighted the importance of recognizing thrombotic risks associated with cisplatin administration and the need to detect them early through imaging. Prompt intervention in such cases is crucial to preventing severe neurological complications.

In summary, the reviewed studies highlight a significant association between moderate to severe headaches and cancer, particularly in patients undergoing chemotherapy.

### 3.9. Systematic Epidemiological Data from Cohort Studies

Beyond individual case reports, broader epidemiological data indicate that headache and other neurotoxic symptoms in cancer patients vary substantially with demographic and treatment-related factors. Patients between the ages of 45 and 55 years appear disproportionately susceptible to neurotoxicity, a pattern supported by Wolff et al.’s systematic review of 80 placebo-cohort oncology trials (17,968 patients), which found that headache reporting was higher in cohorts with younger median age, better performance status and breast cancer diagnoses [8]. These demographic and regimen-intensity effects are reinforced by a recent cohort, where >80% of patients experienced moderate-to-severe chemotherapy-induced neurotoxicity (including headaches), and risk was strongly influenced by agent type and cumulative exposure [73]. Although there are no large-scale reports focused specifically on CRH, converging evidence supports age, regimen intensity and neuroinflammatory burden as key modifiers of CRH susceptibility.

Prospective quality-of-life research using EORTC instruments demonstrates that symptom burden, including pain and treatment-related adverse effects, correlates with both chemotherapy dose modifications and survival outcomes, even if the specific contribution of headache has not been quantified separately. In the CANTO breast cancer cohort (n = 3079), lower baseline physical functioning and higher fatigue on the EORTC QLQ-C30 were independently associated with higher odds of chemotherapy dose reductions and post-chemotherapy toxicities [74]. At the same time, pooled analyses of EORTC trials have shown that multiple symptom and functioning scales are prognostic for overall survival across tumor sites [75]. Clinical descriptions of chemotherapy-induced neurotoxicity further suggest a bimodal temporal pattern: some treatment-related headaches and encephalopathic syndromes present within hours to a few days of drug administration (for example with intrathecal agents or high-dose ifosfamide) and resolve over days to weeks [4,76], whereas other neurotoxic syndromes emerge subacutely 2–14 days after high-dose methotrexate or cytarabine, often coinciding with myelosuppression and systemic inflammatory stress [77]. These data support a broader spectrum of chemotherapy-induced neurotoxicity that can affect dose intensity and potentially survival, but their precise incidence, temporal profile and impact remain under-characterized, underscoring the need for prospective, headache-focused oncology cohorts.

## 4. Mechanisms of Chemotherapy-Related Headaches

There is a critical need for mechanistic research to identify the processes underlying CRHs and to differentiate them from other headache types. Table 3 summarizes the similarities and differences of CRHs with other headache types. Discovering specific mechanisms and biomarkers is expected to enhance diagnosis, enabling more targeted treatments rather than relying on generalized pain management [61,78,79]. The mechanisms involved in CRHs are divided into two general groups: iatrogenic and psychogenic (Figure 2).

### 4.1. Iatrogenic Mechanisms of CRHs

Among the key iatrogenic factors contributing to CRHs, infection and inflammation associated with immune system suppression, BBB impairments, hormonal fluctuations, homeostatic changes, and systemic toxicity have been documented. Figure 2 summarizes mechanisms and pathways contributing to CRH pathogenesis, distinguishing them from general chemotherapy neurotoxicity, which causes also other neurological effects [3,29,36,54] (Figure 2).

Immunosuppression and infections: Chemotherapy weakens immune defenses and induces cytopenia, creating a vulnerability window for opportunistic infections that may contribute to headaches [80]. Neutropenia caused by chemotherapy, which is characterized by a reduction in neutrophils, is particularly hazardous to the body’s ability to fight infections [80,81,82]. Patients undergoing chemotherapy are particularly prone to bacterial, viral, and fungal infections, some of which can lead to meningitis or encephalitis, both of which are known to cause persistent headaches [83,84]. Additionally, IT chemotherapy, which involves administering drugs into the CSF through a lumbar puncture or lateral ventricles, increases the risk of introducing pathogens. This procedure may result in aseptic or infectious meningitis, which is often accompanied by severe headaches. Occasionally, oncologists may need to reduce chemotherapy doses to mitigate infection risks, but this can compromise the relative dose intensity (RDI) and, in turn, treatment effectiveness [85,86]. Finally, latent viruses such as herpes simplex and cytomegalovirus can reactivate during chemotherapy, presenting neurological symptoms, including headaches.Neuroinflammation: Neuroinflammation represents a central convergent pathway in CRH pathogenesis, distinct from the general inflammatory response seen in other chemotherapy complications. Specifically, CRHs exhibit elevated levels of trigeminal-specific inflammatory mediators and preferentially activate pain-processing neural circuits. Neuroinflammation is the second leading mechanism in chemotherapy-induced headaches, driven by the brain’s immune response to tissue damage, toxins, and chemotherapy agents. The inflammation may persist even after the initial trigger is removed, leading to chronic headache syndromes [87,88]. Chemotherapy activates glial cells, such as microglia and astrocytes, which release pro-inflammatory cytokines, including IL-6, TNF-α, and IL-1β, sensitizing pain pathways and contributing to headache development [1,89,90,91]. Although chemotherapy agents are known to trigger CRH through the aforementioned mechanisms, there is no evidence for a role in selective meningeal inflammatory pathology. Indirectly, these agents can trigger a general inflammatory cascade, which leads to meningeal inflammation and irritation. For example, oxaliplatin activates the PI3K-mTOR pathway, which increases cytokine expression and may promote neuroinflammation and headaches [3,89]. Similarly, platinum-based drugs (cisplatin, oxaliplatin), alkylating agents, and taxanes disrupt the neuroimmune balance, causing interference with axonal transport, alterations in ion channels, and mitochondrial dysfunction, all of which exacerbate neuroinflammation [92]. These disruptions increase neuronal excitability and lower pain thresholds, worsening headaches. Cytokine-mediated neuroinflammation associated with chemotherapy has previously been established, although no dedicated biomarker studies have yet quantified IL-6 or TNF-α specifically in patients with CRHs. Work in cancer-related pain and primary headache disorders consistently demonstrates higher serum levels of IL-6 and TNF-α in migraine and chronic migraine compared with controls, with positive correlations between cytokine levels and headache frequency or intensity [93,94]. Chemotherapy-induced neuroinflammation can also alter serotonin signaling, increasing its release through the activation of calcitonin gene-related peptide (CGRP) pathways, both of which are strongly implicated in migraine pathophysiology [3,95]. The rise of CGRP along meningeal vessels contributes to vasodilation and sensitization of trigeminal innervation, further linking chemotherapy to migraine-like headaches [95,96,97,98].Disruption of the blood–brain barrier: BBB is a semipermeable membrane of the vascular system of the CNS that protects the neural tissue from toxins and infection, regulates ion balance and neurotransmitter levels, and transports nutrients and waste between the blood and brain [99,100]. Some chemotherapeutic agents are known to disrupt the integrity of the BBB, increasing its permeability to toxins, inflammatory cytokines, and immune molecules. Drugs such as temozolomide, nelarabine, rituximab, methotrexate and others are known to cross the BBB, contributing to headache development [99,101,102]. For example, in neuro-oncology, protocols that combine intra-arterial hyperosmolar mannitol with methotrexate-based or other intra-arterial chemotherapy regimens produce marked, territory-specific increases in contrast leakage on MRI, consistent with transient BBB opening and enhanced drug penetration into brain tissue [103]. This anatomically specific BBB compromise may explain why headaches predominate over other cognitive symptoms in some patients. The disruption of the BBB allows pro-inflammatory cytokines, such as IL-6 and TNF-α, to penetrate the CNS, leading to heightened pain sensitivity and neuroinflammation [104]. This inflammatory cascade sensitizes the trigeminal nerve, which is closely linked to migraine pathogenesis and may also be involved in chemotherapy-related headaches [105]. Additionally, increased BBB permeability exposes the CNS to chemotherapeutic agents, exacerbating neurotoxicity, neuronal damage, and functional impairments.Homeostatic dysregulations: The systemic toxicity of chemotherapy with homeostatic impairments is another significant contributor to CRHs, as it can disrupt multiple organ systems, including the kidneys, liver, heart, and gastrointestinal tract, causing homeostatic disruptions [106,107,108]. Chemotherapeutic agents, such as cisplatin, cause electrolyte imbalances, leading to hypomagnesemia and hypokalemia, as well as sodium-potassium disturbances and dehydration, which are well-recognized triggers of headaches [109]. Specifically, hypomagnesemia occurs in 40–90% of cisplatin-treated patients, and headache is a recognized symptom of clinically significant magnesium deficiency [110]. Migraine data suggest that low magnesium markedly increases migraine attacks [111], but CRH-related hypomagnesemia has not yet been quantified. Dehydration is particularly relevant in headache pathogenesis, as many chemotherapy patients experience fluid loss due to vomiting and gastrointestinal dysfunction [112]. Dehydration also reduces CSF volume, leading to changes in intracranial pressure that can trigger headaches [113]. Additionally, drugs like methotrexate and cisplatin impair kidney function, further exacerbating fluid and electrolyte imbalances, which contribute to headache severity [114].Other iatrogenic mechanisms contributing to headaches: Hormonal fluctuations may also play a role in CRHs, as some chemotherapies affect the endocrine system, notably cortisol and estrogen levels, both of which are linked to headache disorders. Elevated cortisol levels in chemotherapy patients have been associated with increased headache intensity, similar to patterns observed in migraine [115]. In females, for instance, chemotherapy-induced ovarian dysfunction results in estrogen depletion, a known migraine trigger that may worsen CRHs [115,116,117]. Prospective studies in premenopausal breast cancer patients undergoing chemotherapy demonstrate that over 80% experience chemotherapy-induced amenorrhea with corresponding profound estrogen level drop compared to pre-treatment levels [118]. This precipitous decline correlates with new-onset or worsening headaches in 9–42% in post-menopausal women, with temporal patterns resembling menstrual migraine [119]. However, no study has yet directly correlated serial estradiol measurements with chemotherapy-related headache frequency, so extrapolation from migraine and menopause data remains inferential rather than definitive. Finally, vascular changes induced by chemotherapies, such as cisplatin, can cause vasospasms, endothelial damage, and altered cerebral blood flow, leading to fluctuations in intracranial pressure and headaches [39,120].

### 4.2. Psychogenic Mechanisms Underlying CRHs

In addition to iatrogenic mechanisms, psychogenic factors may play a significant role in the development and exacerbation of CRHs (Figure 2). Cancer treatment is a physically and emotionally taxing process, often leading to mental distress, fatigue, and lifestyle disruptions, all of which contribute to headache susceptibility. While these factors are not direct causes of CRHs, they may amplify pain perception and exacerbate headache intensity, making effective management even more challenging.

Sleep disruptions and fatigue: Sleep disturbances and fatigue are common among cancer patients, both as a direct result of chemotherapy and due to the emotional stress associated with a cancer diagnosis [121]. Chemotherapy-induced insomnia, frequent nocturnal awakenings, and poor sleep quality can lead to increased headache frequency and severity [122]. Disruptions in the circadian rhythm further compound this process, as inadequate sleep can lower pain thresholds and heighten neuroinflammatory responses, making headaches more persistent [123]. Polysomnography-based studies in cancer populations show that objective sleep continuity and architecture can be altered, including more awakenings/arousals and, in some cohorts, reduced REM and slow-wave sleep. However, findings vary by cancer type, disease stage, and treatment status [124]. Experimental evidence also links cytotoxic chemotherapy to sleep fragmentation, along with inflammatory signaling, resulting in a correlation with hypothalamic IL-6 expression [125]. Additionally, chemotherapy often causes profound fatigue, leaving patients in a state of chronic exhaustion. This persistent fatigue is linked to mitochondrial dysfunction, oxidative stress, and neuroinflammatory activation, all of which can contribute to headache pathogenesis. The combination of sleep deprivation and systemic fatigue creates a cycle in which exhaustion and neuroinflammation lead to more severe headaches [126].Anxiety and emotional stress: Anxiety is another well-recognized trigger for headaches. Cancer patients undergoing chemotherapy frequently experience heightened levels of stress and anxiety related to their prognosis, treatment side effects, and financial or social burdens. Studies have shown that individuals with high anxiety levels are more prone to developing chronic headaches, including migraine-like symptoms [127,128,129]. Anxiety triggers autonomic nervous system dysregulation, increasing cortisol and adrenaline release, both of which can exacerbate neuroinflammation, vasoconstriction, and pain perception [130]. Prolonged exposure to high stress can increase muscle tension, particularly in the neck and scalp, contributing to tension-type headaches [131], which are commonly reported in chemotherapy patients. Furthermore, chronic stress can alter neurotransmitter balance, especially in dopaminergic and serotonergic pathways, which are known to play a role in pain modulation and the development of headaches [130]. These neurochemical changes may explain why chemotherapy patients with preexisting anxiety or depressive disorders often experience more frequent and severe headaches compared to those without psychological distress.Disruption of daily routine and social life: Cancer treatment can significantly disrupt daily routines, social interactions, and overall quality of life. Many patients experience isolation, reduced physical activity, and a lack of control over their schedules, all of which contribute to emotional distress and headache exacerbation [132]. Routine disruptions can interfere with meal schedules, hydration habits, and medication adherence, critical factors in headache management. For example, inconsistent eating patterns and dehydration can contribute to hypoglycemia and electrolyte imbalances, known headache triggers [133,134]. Likewise, a lack of structured physical activity can lead to musculoskeletal tension and poor circulation, further contributing to headache development. Additionally, chemotherapy-related nausea, dizziness, and gastrointestinal disturbances can prevent patients from engaging in everyday social and occupational activities, leading to emotional distress and an increased perception of pain [135,136].Neuropathic pain-associated exhaustion leading to CRHs: Chemotherapy-induced neuropathic pain is another major factor that contributes to headache development and worsening fatigue. Many chemotherapeutic agents cause direct nerve damage, leading to burning, tingling, or electric-shock-like pain in the extremities. While neuropathic pain primarily affects the peripheral nervous system, it can also sensitize central pain pathways, making patients more susceptible to headaches [137]. Moreover, prolonged exposure to neuropathic pain drains mental and physical energy, leading to a state of exhaustion and increased pain sensitivity. Patients with persistent neuropathic symptoms often report higher levels of stress, irritability, and sleep disturbances, all of which further exacerbate CRHs [3].Other psychological factors: Beyond sleep disturbances, anxiety, routine disruptions, and neuropathic pain, other psychological factors can significantly influence the severity and persistence of chemotherapy-related headaches. Depression and emotional distress are particularly relevant, as cancer-related depression can alter pain perception, inflammatory responses, and neurotransmitter regulation [138], making headaches more intense and lasting. The psychological burden of undergoing chemotherapy, dealing with uncertainty, and facing physical changes often leads to increased emotional strain, which in turn exacerbates headache symptoms.

Cognitive overload is another potential trigger (Figure 2). Many patients undergoing chemotherapy experience cognitive dysfunction, often referred to as “chemo brain” [139,140,141], which makes it more challenging to manage pain effectively, adhere to treatment regimens, and engage in stress-reducing activities. The cognitive impairments create additional frustration and mental fatigue, further intensifying headaches. Patients struggling with cognitive dysfunction may also have difficulty recognizing early headache triggers and taking preventive measures, leading to more prolonged and more painful episodes. For some cancer patients, post-traumatic stress disorder (PTSD) related to their diagnosis of cancer can manifest as heightened autonomic reactivity, persistent stress, and tension headaches. The psychological trauma of a life-threatening illness, combined with the physical toll of chemotherapy, can lead to long-term neurological effects that contribute to recurrent headaches. Patients experiencing PTSD may also have difficulty sleeping [142,143], maintaining social relationships, or engaging in activities that would typically provide relief from stress, all of which further exacerbate CRHs.

## 5. Management Strategies for Chemotherapy-Related Headaches

Current management approaches require critical evaluation based on evidence quality, strength of recommendations, and specific relevance to CRHs rather than extrapolation from primary headache disorders. Most existing treatment recommendations lack rigorous validation in cancer populations, representing a significant gap in evidence-based care. As they can severely impact quality of life, treatment adherence, and overall well-being, CRHs pose a substantial challenge to the management and care of patients with cancer. Despite the prevalence of CRHs, current management strategies primarily focus on symptomatic relief rather than addressing the underlying mechanisms [26,66,69]. The lack of targeted treatments results in suboptimal headache control, leaving many patients with persistent or recurrent symptoms.

### 5.1. Current Strategies

The management of CRHs typically involves a combination of pharmacological and non-pharmacological approaches to reduce headache frequency and severity (Figure 3). However, the data supporting these interventions vary considerably, and clinicians must understand the strength of evidence when making treatment decisions. We categorize current strategies by level of evidence as follows:

Pharmacological interventions with Level 1 Evidence (supported by randomized controlled trials in cancer populations): Adequate hydration is a well-established standard of care for patients receiving cisplatin-based chemotherapy to reduce nephrotoxicity and electrolyte disturbances, with improved hydration associated with reduced treatment-related symptoms, including headache. A review of randomized trials consistently supports aggressive intravenous hydration protocols (in a few hours rather than days) in cisplatin regimens. However, headache has typically been reported as a secondary or adverse-event outcome rather than a primary endpoint [144,145]. Direct RCTs powered specifically for headache outcomes in this setting have not been published, but hydration remains a biologically plausible and widely endorsed supportive intervention.

Pharmacological interventions with Level 2 Evidence (supported by observational studies or extrapolated from migraine literature): Over-the-counter (OTC) analgesics, such as paracetamol and NSAIDs like ibuprofen or naproxen, are commonly recommended for mild to moderate headaches. Due to the superior anti-inflammatory properties of NSAIDS compared to paracetamol, it is likely to produce a better response, but side effects may limit its use. Critical safety considerations for NSAIDs in chemotherapy patients include significantly increased risk of gastric irritation and ulceration (8–12% incidence vs. 2–4% in general population), renal dysfunction particularly with concurrent nephrotoxic agents like cisplatin (4–7% acute kidney injury incidence), and cardiovascular complications including hypertension and thrombotic events (2–5% incidence) [146]. These risks necessitate careful patient selection and often limit NSAID use to short-term administration.

For patients experiencing migraine-like headaches, clinicians may prescribe triptans, such as sumatriptan or rizatriptan, which act on serotonin receptors to alleviate vascular headaches. Triptans have shown promising results in selected cancer-associated headache settings, such as locally invasive head and neck cancer-related headache relieved by oral sumatriptan [147], and headache in meningeal carcinomatosis responding to rizatriptan [148]. However, no large-scale randomized controlled trials have validated the use of triptans specifically for CRHs, and all current recommendations are extrapolated from their established efficacy in primary migraine. Contraindications including uncontrolled hypertension, coronary artery disease, and concurrent use of specific chemotherapy agents require careful screening before triptan prescription.

For patients whose headaches are linked to neuroinflammation or chemotherapy-induced neuropathic pain, corticosteroids like dexamethasone may be used to reduce inflammation and suppress immune-mediated pain responses [4,149]. Short-term corticosteroids, particularly dexamethasone when administered intrathecally with methotrexate, not only improved symptoms such as headache but also overall survival across multiple cancer groups [150].

Additionally, some patients may benefit from neuropathic pain medications, including gabapentinoids (gabapentin, pregabalin) and serotonin-norepinephrine reuptake inhibitors (SNRIs) like duloxetine, which help modulate pain perception [151,152]. These agents fall into the Level 3 Evidence category (mechanistic rationale with limited clinical data specific to CRHs), though they demonstrate established efficacy for chemotherapy-induced peripheral neuropathy. The theoretical justification for their use in CRHs stems from shared neuroinflammatory and sensitization mechanisms, but dedicated clinical trials evaluating efficacy specifically for headache outcomes are lacking.

Emerging therapies under investigation (Level 4 Evidence–investigational): Calcitonin gene-related peptide (CGRP) pathway inhibitors, including monoclonal antibodies (erenumab, fremanezumab, galcanezumab, eptinezumab) and small-molecule CGRP receptor antagonists (“gepants”), are effective migraine therapies and represent a plausible but unproven option for migraine-like headache phenotypes arising during cancer care. The rationale for exploring these agents in oncology derives from CGRP’s established role in trigeminovascular activation and meningeal neuroinflammation in migraine models [153]. In chemotherapy neurotoxicity models, CGRP signaling has also been implicated in pain sensitization (e.g., paclitaxel-induced neuropathic pain attenuated by CGRP receptor antagonism), supporting broader mechanistic overlap with nociceptive neuroinflammation [154]. However, there is no conclusive clinical evidence for amelioration of CRHs by CGRP, with current discussions extrapolated from their migraine efficacy and pain biology. Prospective trials are therefore needed to determine efficacy, optimal timing relative to chemotherapy cycles, safety in immunocompromised patients, and drug–drug interaction considerations.

Non-pharmacological interventions are also widely recommended, particularly for patients who experience chronic or recurring headaches. Hydration therapy, which involves increasing fluid intake through oral hydration or intravenous fluids, is often used to address dehydration-related headaches, especially in patients receiving nephrotoxic chemotherapy agents such as cisplatin and methotrexate [145,155]. Cognitive-behavioral therapy (CBT) and mindfulness-based stress reduction (MBSR) have also been explored as complementary strategies to help patients manage stress, anxiety, and pain perception associated with CRHs [156]. Lifestyle modifications play a crucial role in preventing headaches. Patients are advised to maintain consistent sleep patterns, practice relaxation techniques, and engage in light physical activity to mitigate stress-related headache triggers. Specifically, sleep hygiene interventions, including consistent sleep–wake schedules, limiting daytime napping to <30 min, and creating optimal sleep environments (dark, quiet, cool), demonstrate a considerable reduction in headache frequency in the cancer cohort [124]. Dietary adjustments to exclude food that triggers migraines (e.g., caffeine, processed meats, and artificial sweeteners) may help manage headache symptoms [26] (Figure 3). However, evidence for dietary modifications in CRHs is sparse, with most recommendations extrapolated from migraine prevention protocols.

### 5.2. Limitations of Current Strategies

Current management strategies for CRHs have several limitations. The most significant challenge is that available treatments focus on symptom relief rather than targeting the underlying mechanisms of chemotherapy-induced headaches [4,9]. General analgesics and triptans may provide temporary relief but do not address neuroinflammation, BBB disruption, or systemic toxicity, which are key contributors to CRHs. Furthermore, treatment failure rates are substantial, and this may depend on perpetuating factors such as drug-related side effects, age or co-morbidities. This leaves a significant unmet medical need for more effective and better-tolerated interventions.

Additionally, many pharmacological options come with potential side effects, particularly in cancer patients who are already dealing with multiple treatment-related complications. NSAIDs, for example, pose risks of gastric irritation, kidney dysfunction, and cardiovascular complications, especially in patients receiving nephrotoxic chemotherapy agents [129]. A meta-analysis of retrospective studies found that concomitant NSAID use is a risk factor for cisplatin-induced nephrotoxicity, supporting avoidance or cautious use of NSAIDs during cisplatin therapy [157]. These risks often require gastroprotective agents such as proton pump inhibitors and careful renal function monitoring, adding complexity and cost to treatment regimens. Similarly, corticosteroids, while effective in reducing inflammation, carry risks of immunosuppression, weight gain, and mood disturbances, which can further complicate cancer treatment. Observational cancer data also show that high-dose corticosteroid exposure is associated with an increased risk of serious infections/hospitalization for infection in oncology populations, underscoring the need for careful risk–benefit assessment and monitoring, particularly when combined with other immunosuppressive therapies [158].

Another major limitation is the lack of standardized treatment and protocols specifically for CRHs. Unlike primary headache disorders, which benefit from evidence-based guidelines from organizations such as the American Headache Society and the European Headache Federation, no major oncology or neurology organization has published comprehensive guidelines for CRH management. The variability in their clinical presentation, ranging from mild discomfort to severe debilitating headaches, complicates treatment selection, as different patients may respond differently to the same interventions [26,66,69]. This heterogeneity likely reflects diverse underlying mechanisms (vascular vs. neuroinflammatory vs. meningeal irritation), suggesting that future treatment algorithms may need to incorporate phenotyping or biomarker-guided approaches to match patients with optimal therapies.

Non-pharmacological approaches, though beneficial, are often underutilized due to a lack of accessibility and patient adherence issues. Hydration therapy, for instance, requires careful monitoring to prevent fluid overload, particularly in patients with compromised kidney function [145]. Patients with reduced renal function or uncompensated cardiac disease often need modified cisplatin hydration (or alternative regimens) because standard saline loading can be contraindicated and requires careful volume-status assessment to avoid fluid overload/pulmonary edema. This limitation is clinically relevant because chronic kidney disease is common in oncology populations, with extensive database analyses reporting CKD prevalence that increases with age (e.g., ~7.9% under 60 vs. ~16.2% over 60 in one large dataset) [159]. Psychological interventions such as CBT and MBSR also require trained providers and program infrastructure, and surveys of cancer survivorship care note accessibility barriers and uneven availability, including challenges in rural settings [160]. Finally, lifestyle and behavioral modifications can be demanding to sustain during periods of severe fatigue, nausea, or cognitive burden, which creates a practical “catch-22” where symptom severity undermines adherence, arguing for simplified, low-effort strategies deliverable even during high-toxicity phases.

## 6. Integrating CRHs into Standard Oncology Care: Practical Recommendations

To address the gap between recognizing CRHs as a significant clinical problem and implementing systematic approaches to their management, we propose evidence-based integration strategies that can be adopted within existing oncology care frameworks. These recommendations are designed to be practical and actionable, drawing on successful implementation models from leading cancer centers while acknowledging the varying resource constraints across healthcare settings. The goal is to shift CRH management from reactive symptom treatment to proactive screening, prevention, and personalized intervention.

### 6.1. Screening and Assessment Protocols

Systematic screening for headache symptoms and risk factors in patients undergoing chemotherapy is increasingly recognized as an essential component of supportive cancer care. Yet, headaches in oncology practice are often identified only after the patient reports significant symptoms. Observational studies note that under-reporting of treatment-related symptoms is common unless clinicians proactively inquire, contributing to delayed recognition and management of headache and other neurotoxic effects [161]. A proactive approach begins with a baseline assessment before chemotherapy initiation. Clinical headache guidelines recommend documenting pre-existing headache disorders using validated instruments such as the Migraine Disability Assessment Scale (MIDAS) and the Headache Impact Test-6 (HIT-6), both of which are brief, patient-reported tools that quantify functional impairment and establish a baseline for longitudinal comparison [162]. These tools are widely used in headache research and clinical practice and are feasible for incorporation into oncology consultations. Beyond questionnaire data, clinical teams should specifically identify high-risk patient characteristics, including younger age, female sex, prior migraine or tension-type headache history, and comorbid anxiety or depression, and plan treatment with high-risk regimens such as platinum-based combinations or intrathecal chemotherapy. This risk stratification allows targeted preventive interventions and heightened surveillance for the highest-risk populations.

Patient education represents another critical component of baseline assessment. Studies on symptom monitoring in oncology demonstrate that patients frequently fail to report adverse symptoms unless they are explicitly counselled on what to expect and when to alert their care team [161]. Educational materials that describe headache as a potential chemotherapy-related symptom and outline typical timing and warning signs may therefore improve early reporting. The use of headache diaries, either paper-based or via smartphone applications, is recommended to track frequency, severity, triggers, and medication use, and can generate valuable longitudinal data for clinical decision-making [163].

Ongoing monitoring during active chemotherapy should incorporate routine assessment of headache symptoms at each treatment cycle. Headache and pain guidelines support the use of numeric rating scales combined with brief functional impact questions to monitor symptom evolution and response to interventions [164]. Regular assessment facilitates the early identification of emerging headaches, evaluation of treatment effectiveness, and timely consideration of supportive measures or chemotherapy modifications. Importantly, assessment protocols must include screening for red-flag features that suggest secondary causes of headache requiring urgent evaluation. Neurology and oncology guidelines emphasize that sudden, severe headache; progressive worsening over days to weeks; focal neurological deficits; altered mental status; fever with neck stiffness; or papilledema warrant prompt investigation to exclude intracranial metastases, meningitis, intracranial hemorrhage, or other serious complications [165].

Neuroimaging decisions should balance diagnostic yield with cost and potential risk. MRI with and without gadolinium contrast is preferred for evaluating new or atypical headaches in cancer patients because of its superior sensitivity for metastases and leptomeningeal disease. At the same time, CT remains appropriate for urgent assessment when MRI is not readily available [166]. In selected patients with persistent severe headaches and unremarkable imaging, lumbar puncture may be indicated to assess cerebrospinal fluid pressure, cell counts, biochemistry, and cytology to exclude infection or leptomeningeal involvement.

### 6.2. Multidisciplinary Care Models

The complexity of CRHs spanning the oncology, neurology, pain medicine, and supportive care domains necessitates coordinated multidisciplinary approaches. Reviews of supportive oncology care emphasize that fragmented symptom management contributes to delayed diagnosis, undertreatment, and unnecessary treatment interruptions, whereas integrated care models improve symptom control and patient experience [167]. International organizations now recommend the early integration of palliative care and pain medicine alongside active cancer treatment, which has been shown to improve symptom burden and quality of life [168]. Within this framework, headache management falls naturally within palliative care expertise in symptom control, quality-of-life optimization, and medication management. Pain medicine specialists bring additional expertise in interventional approaches, such as nerve blocks, trigger-point injections, and advanced pharmacological strategies, including membrane stabilizers and neuromodulatory medications.

At a policy level, the European Society for Medical Oncology (ESMO) recommends systematic assessment of patient-reported symptoms that impact quality of life, including pain and headache, in both clinical trials and routine practice [169]. Successful multidisciplinary care also requires clear communication pathways and defined roles among team members. Oncology nurses often serve as the first point of contact for symptom reporting and, therefore, require education about CRH recognition, appropriate triage decisions, and first-line management strategies. Pharmacists can contribute by conducting medication reviews to identify potential drug–drug interactions, recommending evidence-based analgesic regimens, and counselling patients on appropriate medication use, while social workers and psychologists address the psychosocial contributors to headache, including anxiety, depression, sleep disturbance, and stress management. Physical and occupational therapists may assist with posture optimization, reducing muscle tension, and gentle exercise programs to reduce headache triggers.

### 6.3. Treatment Algorithms and Decision-Making Frameworks

Structured, stepwise algorithms are widely recommended for headache care to guide evaluation, escalation, and exclusion of secondary causes, and similar principles apply in oncology, where headaches may signal serious complications. We propose a severity-based algorithmic approach that can be adapted to individual patient circumstances and institutional resources.

Mild CRHs (pain intensity of 1–3/10 with minimal functional impact): initial management can include simple analgesics (e.g., paracetamol/acetaminophen) and when not contraindicated, short courses of NSAIDs, alongside supportive measures such as adequate hydration (tailored for renal/cardiac comorbidity) and lifestyle measures aligned with headache guidance (regular sleep, meals, stress management, trigger avoidance).

Moderate CRHs (pain intensity of 4–6/10 with moderate impairment): escalation may include prescription NSAIDs where appropriate, with PPI gastroprotection for patients at elevated GI risk and careful renal/cardiovascular assessment. For migraine-like presentations, clinicians may consider a triptan, but only after screening for contraindications such as ischemic heart disease/coronary vasospasm and uncontrolled or significant hypertension. Neurology referral is appropriate for frequent, persistent, atypical, or treatment-limiting headaches, consistent with oncology headache-evaluation guidance.

Severe or refractory headaches (pain intensity of 7–10/10, disabling, or nonresponsive): management should priorities urgent evaluation for secondary causes, including neuroimaging when clinically indicated, because cancer patients are at risk for intracranial metastases, leptomeningeal disease, infection, hemorrhage, and treatment-related complications. Interventional approaches such as greater occipital nerve block have evidence in primary headache disorders (e.g., migraine/cluster) [170], but CRH-specific evidence is lacking; therefore, use should be individualized rather than presented as established for CRHs.

Prevention strategies merit special consideration for patients identified at high risk during baseline screening or who experience significant headaches with initial chemotherapy cycles. Prophylactic approaches may include scheduled rather than as-needed analgesics taken regularly during the expected time window of CRH occurrence based on prior cycle patterns, aggressive prophylactic hydration and electrolyte replacement as discussed above, consideration of preventive medications used in primary headache disorders such as beta-blockers, tricyclic antidepressants, or anticonvulsants, though noting limited specific evidence in CRH populations. In some cases, this may involve discussion with the oncology team regarding chemotherapy dose reduction or the substitution of alternative agents with lower headache risk when oncologically appropriate.

Throughout the implementation of these algorithms, ongoing monitoring and adjustment remain essential. Treatment response should be systematically evaluated using the same headache assessment tools employed for screening, allowing objective measurement of improvement. Patients should be empowered to track their headaches using diaries or electronic tools, recording not only pain intensity but also timing relative to chemotherapy, associated symptoms, medication use and effectiveness, and functional impact. These longitudinal data inform the iterative refinement of management strategies, identify successful approaches to be continued, and flag ineffective interventions to be discontinued or modified. Regular team communication, including brief huddles or structured case review meetings, ensures that headache management remains integrated with overall cancer care rather than occurring in isolation.

## 7. The Way Forward: Toward Effective Management of CRHs

Given the complex interplay of factors in CRHs, a multifaceted approach is essential to improve patient outcomes. To meaningfully advance the field beyond current limitations, we identify priority research initiatives, therapeutic development pathways, and healthcare system changes required to transform CRH management from palliative symptom control to mechanism-targeted disease-modifying treatment.

### 7.1. Priority Research Initiatives That Would Most Advance the Field

The most urgent need is for high-quality prospective research to establish the true epidemiology, natural history, and risk factors for CRHs across diverse cancer types and treatment regimens. Specifically, a multicenter international prospective cohort study enrolling at least 500 patients receiving standard chemotherapy protocols, with systematic headache assessment using validated instruments at baseline and throughout treatment, would provide definitive incidence data, identify modifiable risk factors, characterize temporal patterns, and lay the foundation for prevention trials. This cohort should include comprehensive biomarker collection, including serial measurements of inflammatory cytokines (IL-6, TNF-α, IL-1β), CGRP levels, markers of blood–brain barrier disruption, and genetic polymorphisms that may influence drug metabolism or headache susceptibility. Biobanking of plasma and cerebrospinal fluid samples, where ethically feasible, would enable future discoveries in science as new biomarker technologies emerge.

Phase II and III randomized controlled trials of CGRP pathway inhibitors versus standard care represent another high-priority initiative given a strong mechanistic rationale and promising preliminary case report data. A pragmatic trial design comparing erenumab or fremanezumab administered monthly versus usual care in patients receiving platinum-based chemotherapy would test both efficacy for headache prevention and safety in immunocompromised populations. Outcomes should include not only headache frequency and severity but also chemotherapy completion rates, quality-of-life measures, and economic analyses of cost-effectiveness. If proven effective, CGRP inhibitors could rapidly translate into clinical practice given their existing FDA approval for migraine.

Advanced neuroimaging studies characterizing brain network alterations in CRH patients versus chemotherapy patients without headaches would provide mechanistic insights that are currently lacking. Functional MRI studies with standardized headache-provocation paradigms could identify specific neural circuits activated during CRH episodes, whereas resting-state connectivity analyses might reveal network alterations predisposing to headache development. PET imaging using ligands targeting neuroinflammation could quantify glial activation and neuroinflammatory processes hypothesized to contribute to CRHs. Diffusion tensor imaging, which assesses white matter integrity, might detect blood–brain barrier compromise or direct neurotoxic effects. These multimodal imaging studies in carefully stratified patients would generate hypotheses about pathophysiology that are testable in preclinical models.

Genomic and pharmacogenomic studies represent another frontier likely to yield personalized medicine approaches. Genome-wide association studies in large patient cohorts could identify genetic variants associated with CRH susceptibility, potentially implicating specific biological pathways as therapeutic targets. Candidate gene studies focusing on polymorphisms in drug-metabolizing enzymes, inflammatory mediators, or pain-processing genes could explain individual variability in CRH risk and inform patient-specific risk prediction. Pharmacogenomic analyses correlating genotype with treatment response would enable precision-matching of patients to optimal therapies, improving efficacy while minimizing trial-and-error prescribing.

### 7.2. Advancing Mechanistic Research Through Translational Models

Research on molecular processes and mechanisms, as well as the identification of biomarkers, is expected to facilitate early detection and effective therapeutic engagement with CRHs [60]. Investigating inflammatory cytokines, CNS neurotransmitter imbalance, and oxidative stress markers could provide insights into the pathophysiology of CRHs, enabling targeted drug development. Additionally, neuroimaging techniques such as functional MRI (fMRI) and PET scans should be explored to elucidate how chemotherapy alters the functional dynamics of the brain and pain-processing pathways [171,172]. By pinpointing specific neural changes associated with CRHs, researchers and clinicians can develop more precise treatment strategies tailored to individual patients.

Translational research bridging preclinical models and human studies remains essential for testing mechanistic hypotheses and screening potential therapeutic agents. Development of animal models that recapitulate key features of CRHs, including meningeal inflammation, blood–brain barrier disruption, and behavioral correlates of headache, would also enable the systematic testing of pathophysiological hypotheses and drug candidates before human trials. However, current animal models have significant limitations as animals cannot directly report headache, requiring reliance on surrogate measures such as facial grimacing, periorbital sensitivity, or avoidance behaviors that may not fully capture the human experience.

### 7.3. Targeted Pharmacological Interventions Based on Mechanism

A shift from conventional painkillers to personalized, more effective and targeted treatments is necessary for improving CRH management. Since neuroinflammation is a critical contributing factor, the efficacy of anti-inflammatory drugs beyond NSAIDs should be systematically investigated. Monoclonal antibodies targeting inflammatory pathways, such as TNF-α inhibitors or IL-6 inhibitors, which have shown promise in inflammatory and autoimmune conditions [173], may be repurposed or modified for CRH management. However, caution is warranted, as systemic immunosuppression in cancer patients carries significant infection risks and may interfere with cancer immunosurveillance, requiring careful safety evaluation in early phase trials before broader implementation. Novel neuroprotective agents, such as mitochondrial enhancers, drugs that protect the BBB, and ion channel modulators, are of increasing interest and warrant research and evaluation in clinical trials [174]. Finally, melatonin, which regulates serotonin, dopamine, and glutamate pathways, is commonly used in the management of migraine and neuropathic pain [174] and may also be beneficial in reducing the severity of CRHs.

### 7.4. Optimizing Supportive Care and Non-Pharmacological Interventions

While pharmacological advancements are crucial, supportive care interventions should not be overlooked. For instance, hydration protocols should be standardized across oncology centers to prevent dehydration-related headaches, particularly among patients receiving nephrotoxic chemotherapies. Electrolyte monitoring and correction should become routine in cancer care, as sodium, potassium, and magnesium imbalances can contribute to headaches [109]. Development of institutional protocols standardizing hydration volume, timing, and electrolyte supplementation according to specific chemotherapy regimens would reduce practice variation and ensure that all patients receive evidence-based preventive care. Notably, CBT, mindfulness-based stress reduction (MBSR), and relaxation techniques have been shown to be effective in reducing the intensity and frequency of chronic headaches [156]. Cancer patients experiencing high levels of anxiety and emotional distress may benefit from psychosocial support [175], which can reduce stress-related headache exacerbation. Expansion of telehealth delivery models for psychological interventions could address geographic disparities in access, allowing rural patients to participate in evidence-based programs remotely. Finally, physical therapy, gentle exercise, and massage therapy may help alleviate tension headaches and improve blood circulation. Patients should be educated on lifestyle modifications, including maintaining good sleep hygiene, avoiding triggers that cause headaches, and adopting stress management techniques to minimize headache frequency [176].

### 7.5. Developing Personalized Treatment Plans Through Precision Medicine

A one-size-fits-all approach is unlikely to be effective in managing CRHs, given the variability in headache presentation among patients. Future treatment should be guided by personalized medicine, informed by mechanistic studies and biomarkers, and consider individual patient risk factors, chemotherapy regimens, genetic predisposition, and comorbid conditions. Using artificial intelligence (AI) and machine learning models in cancer care may help predict which patients are at the highest risk for developing CRHs [61]. Specifically, machine learning algorithms trained on large datasets incorporating clinical variables, genetic data, treatment parameters, and longitudinal symptom reports could generate individualized risk scores to identify high-risk patients before headaches develop, enabling pre-emptive intervention. AI-driven predictive models could also be used to tailor treatment recommendations for specific patients and optimize therapeutic outcomes. These algorithms could integrate real-time data from patient-reported outcome applications to suggest dosage adjustments or alternative medications based on ongoing response patterns, thereby creating dynamic, adaptive treatment protocols.

### 7.6. Clinical Trials Establishing Evidence-Based Guidelines

Currently, there are no standardized treatment guidelines for CRHs. Well-designed clinical trials combining pharmacological interventions with integrative care models are warranted to build a solid scientific ground for evidence-based therapy of CRHs. Specifically, pragmatic randomized trials comparing different management strategies in real-world oncology settings would generate evidence directly applicable to routine practice. Adaptive trial designs that allow mid-trial modifications based on accumulating data could accelerate the identification of effective approaches while maintaining scientific rigor. Healthcare organizations should prioritize developing clinical guidelines that provide clear diagnostic criteria and treatment protocols. Major oncology societies, including the American Society of Clinical Oncology, the European Society for Medical Oncology, and the National Comprehensive Cancer Network, should convene expert panels to develop consensus guidelines for CRH screening, diagnosis, and management based on systematic evidence reviews. These guidelines should explicitly grade recommendations by evidence quality, acknowledge uncertainties where evidence is lacking, and identify priority research questions to guide future investigation. Establishing evidence-based treatment protocols should enable effective, standardized headache management across different medical organizations, cancer types, and patient cohorts.

### 7.7. Healthcare System and Policy Changes Supporting CRH Management

Beyond clinical and research initiatives, healthcare system factors significantly impact CRH management. Reimbursement policies that fail to adequately compensate time-intensive supportive care activities, including symptom assessment, patient education, and care coordination, create financial disincentives for comprehensive CRH management, particularly in resource-constrained settings. Advocacy for appropriate payment models recognizing the value of supportive oncology care is needed. Quality metrics and accreditation standards for cancer programs should explicitly incorporate systematic headache assessment and management, creating accountability for attention to this critical quality-of-life domain. Public reporting of patient-reported outcomes, including headache burden, would provide transparency and incentivize institutional improvement. Finally, patient advocacy organizations can play a crucial role by raising awareness, supporting research funding, and ensuring that patient voices are considered in shaping research priorities and in clinical guideline development.

## 8. General Considerations and Future Directions

The lack of effective CRH management leaves many patients struggling with mild to severe headaches, further diminishing their quality of life. Given the pressing need for disease-modifying therapies and the significant burden imposed by CRHs, more comprehensive research is needed to understand the underlying mechanisms and to develop targeted treatments that go beyond mere symptomatic relief. While NSAIDs or OTC analgesics can offer temporary relief to many patients with CRHs, they fail to address the underlying pathophysiological mechanisms [177]. In severe cases, opioids are prescribed, which come with risks of addiction and side effects that can further compromise a patient’s overall health. Unlike migraines or tension-type headaches, which have established diagnostic guidelines, CRHs are not considered a condition that warrants serious consideration with effective intervention. The lack of definitive biomarkers makes it challenging to differentiate migraine and primary headache disorders in cancer patients from CRHs [178]. This diagnostic ambiguity has practical consequences, including difficulty conducting rigorous clinical trials with well-defined patient populations, a lack of insurance coverage codes specific to CRHs, potentially limiting reimbursement for specialized care, and an absence of quality metrics tracking CRH outcomes in cancer programs. Formal recognition of CRHs within international headache classification systems represents a vital advocacy goal that would legitimize this condition and facilitate research, clinical care, and policy development. Finally, the significant variability in drug response, administration route, and effective dosage of painkillers further complicates efforts to establish a universal treatment approach for CRHs. Considering all of the discussions, the practical advances in the management or treatment of CRHs warrant efforts in three key areas:(1)Mechanistic studies investigating the roles of inflammation, BBB disruption, mitochondrial dysfunction, sensitization, and neurotransmitter imbalances in the CNS of CRHs. Investigating these pathways should help elucidate the pathobiology of CRHs and identify novel therapeutic targets that extend beyond conventional symptomatic relief.(2)Establishing biomarker-guided standardized diagnostic criteria for CRHs and their integration in treating cancer patients and clinical care. These will involve routine assessment of headache patterns in chemotherapy patients, along with preventive steps such as improved hydration, sleep regulation, and customized pain management plans.(3)Multicenter clinical trials with pharmacological interventions combining traditional NSAID and OTC painkillers with triptans, anti-CGRP antibodies, hormone therapies and others, as well as treatment of hypertension, vascular dysfunction, and neural sensitization that might contribute to CRHs. This work must be conducted with careful consideration of potential cross-interactions of drugs with toxicity.

Ramping up research efforts, refining diagnostic approaches, and developing more effective therapeutic strategies are thus essential to alleviating the burden of CRHs from patients undergoing chemotherapy. In combination with non-pharmacological treatment approaches, these efforts are expected to enhance the quality of life for cancer patients, improve their treatment adherence, and ultimately lead to better overall outcomes. As cancer survival continues improving with advances in oncology, the population of cancer survivors dealing with the long-term sequelae of treatment will grow substantially. Understanding and effectively managing CRHs represents not only an acute symptom control issue but also a survivorship care challenge that will become increasingly important in the decades ahead.

## 9. Limitations of the Review

While this review provides a first comprehensive synthesis of existing literature on CRHs, several limitations should be acknowledged. One of the primary challenges is the sparsity of targeted research on CRHs, as much of the existing data on chemotherapy-related neurological effects focuses on more widely recognized conditions such as neuropathy, cognitive dysfunction, and cerebrovascular complications. As a result, much of the discussion in this review originates from studies of broader chemotherapy-induced neurotoxic effects, including headaches. Another limitation is the variability in study methodologies and definitions of CRHs. Many reports do not distinguish between primary headaches (migraines or tension-type headaches) and CRHs in cancer patients, making it difficult to establish clear diagnostic criteria and prevalence estimates (Table 3). The lack of standardized classification and differential diagnosis for CRHs means that different studies report varying frequencies and severities of these headaches, complicating the ability to draw firm conclusions about their epidemiology, potential mechanisms, and risk factors. The review is also constrained by its reliance on retrospective studies, case reports, and small-scale clinical investigations, many of which lack control groups or rigorous methodology to establish causation. The lack of large, randomized controlled trials (RCTs) specifically focused on CRHs limits the strength of the evidence available to determine effective treatment strategies.

Additionally, much of the current literature does not account for potential confounding factors, such as preexisting headache disorders, psychological stress, or concurrent medications, which could influence the reported incidence and severity of CRHs. The lack of diversity in the patient populations studied, with most work focused on specific cancer types or treatment regimens, is likely to introduce selection bias into the understanding and interpretation of CRHs. Given all of the above, the findings reported in this review may not be generalizable to all cancer patients, particularly those receiving combination chemotherapy regimens or undergoing novel targeted therapies that were not widely studied in earlier research. Finally, this review is limited by its reliance on published, peer-reviewed literature, with unpublished data and ongoing clinical trials excluded from the current analysis. Notwithstanding these limitations, the current study highlights significant progress and major knowledge gaps regarding CRHs, underscoring the need for further investigation into their mechanisms, diagnosis, and management.

## 10. Conclusions

CRHs can affect up to 30% of cancer patients receiving chemotherapy. This review advocates CRHs as a distinct clinical entity requiring targeted diagnostic and therapeutic strategies, differentiated from primary headache disorders, intracranial metastases, and procedure-related meningeal irritation through specific temporal relationships, mechanistic pathways, and clinical features outlined in the differential diagnostic framework. Despite alarming numbers and detrimental effects on well-being and treatment outcomes, CRHs have received limited research attention, lacking mechanistic data, standardized diagnostic criteria, and targeted treatment approaches. The existing management strategies primarily rely on symptomatic relief, failing to address the condition’s underlying causes. The absence of mechanistic understanding and targeted treatments continues to leave patients with persistent and often debilitating headaches, negatively impacting their quality of life and treatment adherence.

Throughout this review, we presented the first comprehensive analysis of CRHs as an independent clinical entity deserving focused investigation and systematic management. While some chemotherapeutic agents, such as methotrexate, cisplatin, and oxaliplatin, have been implicated explicitly in headache development, the underlying molecular mechanisms and biological processes of CRHs remain poorly understood. The proposed differential diagnostic framework and integration recommendations provide immediately actionable tools for clinical implementation, enabling oncology teams to screen, diagnose, and manage CRHs systematically. At the same time, prospective research helps validate and refine optimal management strategies. Importantly, the variability in headache presentation across different chemotherapies, along with the lack of standardized approaches, complicates the diagnosis and treatment of CRHs, underscoring the urgent need for personalized management.

With advances in biomarker discovery, neuroimaging tools, and mechanistic studies, we anticipate progress in the timely diagnosis and effective management of CRHs in the foreseeable future. Specifically, the priority research initiatives, including large prospective cohorts with comprehensive biomarker collection, randomized trials of mechanism-targeted therapies such as CGRP inhibitors, and advanced neuroimaging studies, represent clear pathways toward evidence-based precision medicine for CRHs. Combining multidisciplinary care that includes oncologists, neurologists, pain specialists, and supportive care providers should enable better headache management and facilitate the integration of CRHs into cancer treatment protocols. Conversely, addressing CRHs through mechanistic research and disease-modifying treatments will improve chemotherapy outcomes and treatment adherence, with implications for patient well-being and disease prognosis.

Given the growing burden of cancer worldwide, the detrimental effects of CRHs on treatment adherence and quality of life, and improving the prognosis of chemotherapies with better life expectancy, extending the duration during which patients experience treatment-related toxicities, the recognition and treatment of CRHs as one of the principal adverse effects of chemotherapy should become a clinical priority in oncological care. The time has come to move beyond viewing headaches as minor, expected inconveniences of cancer treatment toward recognizing them as clinically significant complications warranting the same systematic attention, research investment, and therapeutic innovation applied to other chemotherapy-induced toxicities such as neuropathy and cognitive impairment. Only through this paradigm shift can we hope to substantially improve outcomes for the millions of cancer patients suffering from this underrecognized yet highly impactful complication.

## Figures and Tables

**Figure 1 ijms-27-00262-f001:**
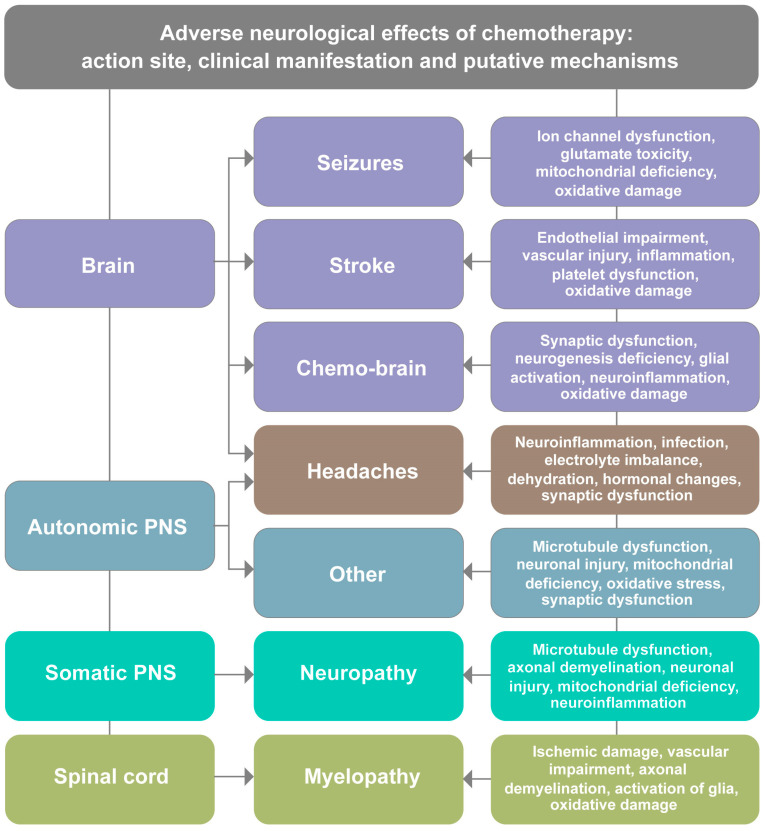
Schematic representation of adverse neurological effects of chemotherapy: action site, clinical manifestation and putative mechanisms. As illustrated, chemotherapy-related headaches are contributed to by effects on the brain and the peripheral autonomic nervous system (PNS). Unlike widely recognized chemotherapy-related clinical conditions such as seizures, stroke, chemo-brain, neuropathy and myelopathy, headaches are largely overlooked and poorly understood.

**Figure 2 ijms-27-00262-f002:**
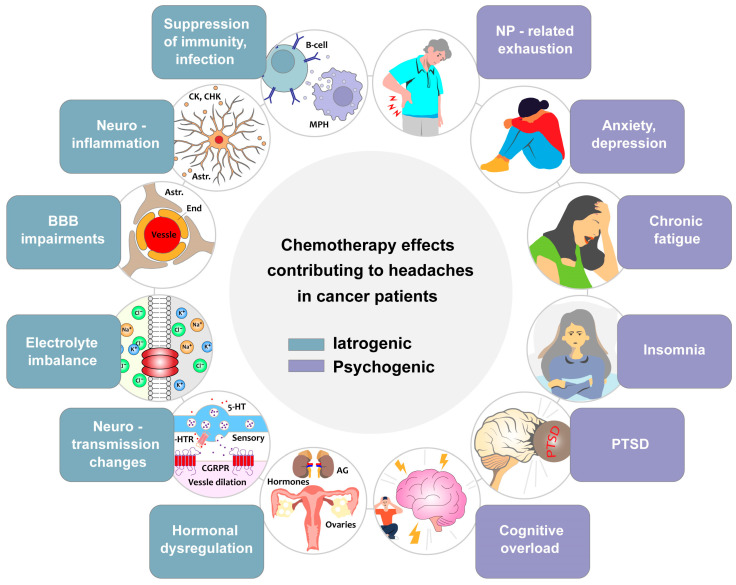
Schematic illustration of chemotherapy effects contributing to headaches. The effects are divided into two groups: (1) Iatrogenic (sage) and (2) Psychogenic (mauve). Abbreviations: MPH—macrophages; CK—cytokines; CHK—chemokines; Astr—astrocytes; End—endothelium; CGRP—calcitonin-gene related peptide; AG—adrenal glands; PTSD post-traumatic stress disorders; BBB—blood–brain barrier; NP—neuropathy.

**Figure 3 ijms-27-00262-f003:**
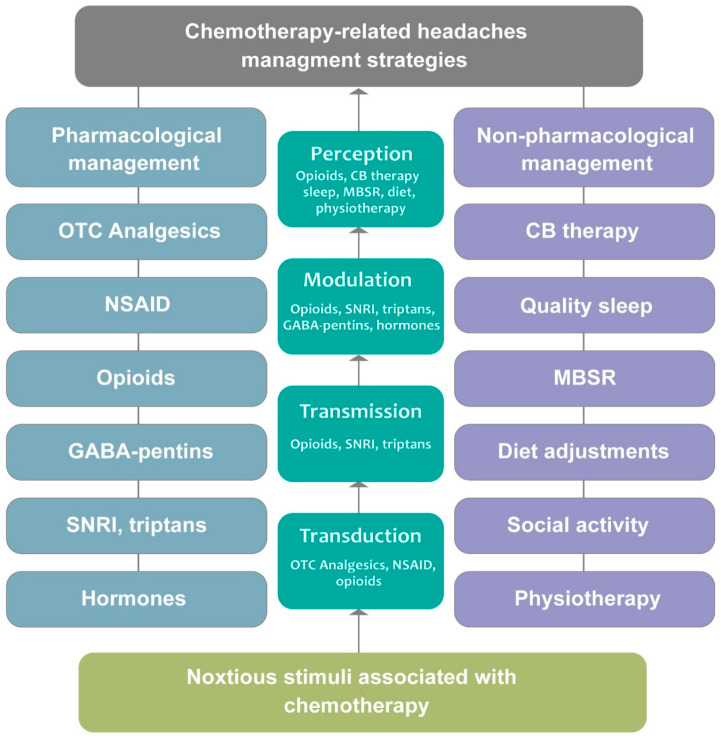
Current approaches used for management—alleviation of CRH and levels of their action along the nociception pathway. The strategies used for management of CRHs are classified into two broad groups: (1) pharmacological and (2) non-pharmacological. While the former interferes with the generation and processing of nociceptive signals at multiple levels, the latter dampens the pain response at the highest level, i.e., the perception stage. Abbreviations: OTC—over the counter; NSAID–non-steroid anti-inflammatory drugs; GABA—γ-aminobutyric acid; SNRI—serotonin-norepinephrine reuptake inhibitors; CB—cognitive behavioral; MBSR—mindfulness-based stress reduction.

**Table 1 ijms-27-00262-t001:** Comparative incidence of chemotherapy-related neurological complications and their co-occurrence with CRHs.

Neurological Complication	Reported Incidence (%)	Co-Occurrence with CRHs (%)	Primary Contributing Agents	Reference
Chemotherapy-Related Headaches	10–30%	N/A	Platinum compounds, methotrexate, cytarabine	[4,8,9]
Peripheral Neuropathy	30–70%	15–25%	Taxanes, platinum compounds, and vinca alkaloids	[10,11]
Cognitive Impairment	20–75%	20–30%	Methotrexate, cisplatin, cyclophosphamide	[12,13,14,15]
Seizures	1–5%	10–15%	Methotrexate, ifosfamide, cytarabine	[16,17,18]
Stroke	1–3%	5–10%	Bevacizumab, cisplatin, 5-fluorouracil	[19,20]
Myelopathy	<1%	Rare	Methotrexate (intrathecal), cisplatin	[21,22]

**Table 2 ijms-27-00262-t002:** Summary of studies reporting the prevalence of headaches and their characteristics in different types of cancer, along with potential mechanisms and their impact on patient care.

Cancer Type	Prevalence of Headaches (%)	Population Demographics	Headache Characteristics	Potential Mechanisms	Impact on Cancer Type	Ref.
Breast Cancer	29.8% (from the study cohort)	Mean age 53.5 years, majority hormone receptor-positive	Varying headache intensity and duration, from dull to migraine-like	Chemotherapy (92% received), endocrine therapy (66%), radiotherapy (52%)	Contributes to treatment burden, affecting quality of life	[66]
B-cell Acute Lymphoblastic Leukemia (ALL)	100% (Single Case)	25-year-old male	Headache, vomiting, disorientation, seizures within 24 h of chemotherapy	Neurotoxic effects of MINE chemotherapy protocol (Ifosfamide, Mitoxantrone, Etoposide)	Chemotherapy was withheld due to neurological symptoms.	[67]
B-cell Acute Lymphoblastic Leukemia (ALL)	100% (Single Case)	2-year-old	Frontal headache for one week, associated with visual disturbances	Methotrexate-induced neurotoxicity, possible CNS involvement	Differential diagnosis expanded to rule out CNS complications	[68]
Chronic Lymphocytic Leukemia (CLL)	100% (Single Case)	56-year-old male	Severe, persistent headaches following chemotherapy	Possible neurotoxicity from chemotherapy	Need for alternative pain management strategies.	[69]
Lymphoblastic Lymphoma	100% Single Case	10-year-old male	Numbness of the face and lips, progressing to systemic pain, including headache	Vincristine-induced neurotoxicity, inflammatory response to chemotherapy	Analgesic pump is used for pain relief	[70]
Non-Hodgkin’s Lymphoma	10.3% (reported headache frequency)	Median age 55 years, 33.3% female	Recurrent headaches lasting 4–36 h (median 15 h)	Intrathecal chemotherapy (methotrexate, cytarabine, hydrocortisone	Pain management necessary during chemotherapy	[71]
Small-Cell Lung Cancer	100% (Single Case)	58-year-old male smoker	Severe, holocranial headache, pressure-like, unresponsive to NSAIDs	Chemotherapy-induced neurotoxicity, intracranial pressure changes	IV morphine is required due to a treatment-resistant headache	[6]
Testicular Cancer	100% (Single Case)	22-year-old male	Moderate, persistent headache, worsened by coughing and straining	Possible vasculotoxic effects of cisplatin leading to cerebral venous sinus thrombosis	Awareness of thrombotic risks is necessary	[72]

**Table 3 ijms-27-00262-t003:** Differential diagnosis of CRHs and treatment implications.

Headache Type	Distinguishing Features	Possible Overlap with CRHs	Primary Diagnostic Considerations	Treatment Implications
Migraine	Unilateral, pulsating pain with nausea, photophobia, and phonophobia	CRHs may mimic migraine features due to neuroinflammation and CGRP pathway activation	Presence of aura, responsiveness to triptans, and history of prior migraines	Triptans, anti-CGRP monoclonal antibodies, and lifestyle modifications
Tension-Type Headache	Bilateral, pressing or tight sensation, mild to moderate intensity	Some CRHs resemble tension headaches due to stress and muscle tension	Absence of nausea and photophobia, linked to psychological stress	NSAIDs, stress management, and muscle relaxation techniques
Meningitis-Related Headache	Severe, persistent headache with fever, neck stiffness, and altered mental status	IT chemotherapy may trigger aseptic or infectious meningitis	Presence of CSF abnormalities on lumbar puncture	Empirical antibiotics for infections, corticosteroids for inflammation
Medication Overuse Headache (Rebound Headache)	Worsening headache with frequent analgesic use	Long-term use of NSAIDs or triptans in CRH patients may cause rebound effects	Resolves upon medication withdrawal	Gradual discontinuation of overused medications, preventive therapy
Hypertensive Headache	Headache with marked blood pressure elevations, often occipital	Specific chemotherapy agents (e.g., bevacizumab) increase blood pressure	Presence of hypertension on clinical evaluation	Anti-hypertensive therapy, lifestyle modifications
Cerebrovascular Headache (Stroke-Related)	Sudden-onset, severe headache with neurological deficits	Chemotherapy increases stroke risk due to vascular toxicity	MRI or CT scan to assess ischemic or hemorrhagic events	Immediate stroke management, anticoagulation if indicated

## Data Availability

This study is a review article and does not report any new data. All data discussed in this manuscript are available in the cited primary sources and published literature.

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
