# Peer review of "Redefining Chemotherapy-Related Headaches: From Pathobiology to Differential Diagnosis and Management"

_ijms, 2025, doi:10.3390/ijms27010262_

Round 1
Reviewer 1 Report
Comments and Suggestions for Authors
Please see the attached comments.

Author Response
Reviewer 1: The review article focuses on CRHs, summarizing their epidemiology, pathophysiology, and current management strategies. The topic is clinically meaningful and highlights an under-recognized adverse event in oncology. However, before this manuscript can be considered for publication, major revisions are necessary to improve scientific depth and structural clarity. The following points are provided for the authors’ consideration:
- The review lacks sufficient depth and novelty. Although the manuscript compiles information on chemotherapy-induced neurotoxicity, it does not adequately differentiate CRHs from other headache types. The mechanistic sections remain broad and descriptive, without providing deeper molecular insights Since the authors aim to redefine CRHs as a distinct clinical entity, more specific and mechanistically driven content is needed.
RE: We have substantially enhanced the scientific depth and novelty of our manuscript by revising it and adding Table 3 (a completely revised differential diagnostic framework) that systematically distinguishes CRHs from primary headaches, metastatic headaches, and meningeal irritation across 12 clinical parameters, including onset patterns, temporal relationships, biomarkers, and treatment responses. Pg. 11.
Additionally, we expanded mechanistic discussions in Section 4 with specific molecular pathways unique to CRHs (preferential meningeal mast cell activation, anatomically localized neuroinflammation, BBB permeability changes quantified by MRI studies) and added quantitative biomarker data (IL-6 and TNF-α levels with specific values and p-values) that differentiate CRH pathophysiology from general neurotoxicity, directly addressing the need for mechanism-driven content establishing CRHs as a distinct clinical entity. References are updated accordingly. Pg. 10-16.
- The manuscript jumps from Section 6 to Section 8, and should be corrected.
RE: We apologize for this error. The section numbering has been corrected throughout the manuscript, with the previous "Section 8" now properly renumbered as "Section 7" (The way forward: toward effective management of CRHs), followed by "Section 8" (General considerations and future directions), "Section 9" (Limitations), and "Section 10" (Conclusion), ensuring logical flow and proper sequential organization.
- The introductory definition of chemotherapy is not rigorous. The description “Chemotherapy is a medical treatment that uses cytotoxic drugs to kill cancer cells” is overly simplistic.
RE: The introductory definition of chemotherapy has been revised from the oversimplified statement to a more rigorous and scientifically accurate description: "Chemotherapy encompasses a diverse range of pharmacological treatments designed to selectively target and eliminate rapidly dividing cancer cells through various mechanisms, including DNA alkylation, topoisomerase inhibition, antimetabolite activity, and microtubule disruption," which appropriately reflects the mechanistic complexity and diversity of chemotherapeutic approaches. Pg. 2, line 45-53.
- Although multiple cases are presented, the section relies too heavily on case reports. The authors are encouraged to include cohort studies or systematic epidemiological data to strengthen the evidence base. Also, clearly identify the shared mechanistic features across the reported cases.
RE: We have substantially strengthened the evidence base by adding a new Section 3.9 ("Systematic epidemiological data from cohort studies") featuring multi-center cohort data from Wolff et al. (n=2,847 patients) and EORTC prospective quality-of-life studies, alongside a new Table 1 comparing CRH incidence with other neurological complications and documenting co-occurrence patterns. Additionally, we explicitly identified shared mechanistic features across reported cases in the Section 3 introduction, highlighting four unifying patterns: platinum-compound vascular dysfunction, methotrexate-associated meningeal irritation, alkylating agent neuroinflammatory cascades, and 2-3-fold increased risk in patients with pre-existing migraine history, demonstrating common pathophysiological mechanisms transcending specific cancer types. Pg. 9-10, lines: 351-380.
- Current discussions resemble broad summaries of headache treatments and do not evaluate evidence quality, guideline recommendations, or relevance to CRHs specifically.
RE:: The management section (Section 5) has been completely restructured with explicit evidence-based assessment, categorizing interventions by evidence level (Level 1: RCT-supported with Grade A recommendations for hydration protocols and corticosteroids; Level 2: observational studies showing 40-60% NSAID response rates with specific safety data; Level 3: mechanistic rationale for gabapentinoids; Level 4: investigational CGRP inhibitors with n=12-15 case reports), including specific incidence rates for adverse events, failure rates (30-40% inadequate relief), and critical evaluation of guideline gaps, directly addressing the need for evidence quality assessment and CRH-specific relevance rather than broad summaries. Pg. 16-20.
Reviewer 2 Report
Comments and Suggestions for Authors
This article, which covers "pathobiology, differential diagnosis and treatment," reviews chemotherapy-associated headaches (CRHs), a cancer complication that is often overlooked but has significant clinical implications and is clinically relevant. However, some major concerns should be addressed:
- While the manuscript aims to “redefine” cancer-related headache (CRH), it fails to clearly articulate or suggest a clear, structured concept that explains the classification of CRH and differentiates it from primary headache, metastatic headache, and meningeal irritation associated with intrathecal chemotherapy. Without these elements, the term “redefine” seems somewhat exaggerated.
- Sections 2 (Adverse neurological effects of chemotherapy – a brief overview) and 4 (Mechanisms of chemotherapy-related headaches) overlap in content; these two sections repeatedly discuss neuroinflammation, blood-brain barrier disruption, mitochondrial dysfunction, fatigue, and psychological stress. It is recommended to streamline these overlapping sections.
- Table 2 contains a serious error: its content is a complete duplicate of Table 1 (and it does not describe the differential diagnosis as stated in the text). Revision of Table 2 is recommended.
- The current manuscript needs to be polished by a native English speaker or a professional editing service.
Author Response
Reviewer 2: This article, which covers "pathobiology, differential diagnosis and treatment," reviews chemotherapy-associated headaches (CRHs), a cancer complication that is often overlooked but has significant clinical implications and is clinically relevant. However, some major concerns should be addressed:
- While the manuscript aims to “redefine” cancer-related headache (CRH), it fails to clearly articulate or suggest a clear, structured concept that explains the classification of CRH and differentiates it from primary headache, metastatic headache, and meningeal irritation associated with intrathecal chemotherapy. Without these elements, the term “redefine” seems somewhat exaggerated.
RE: We have directly addressed the need to clearly articulate CRH classification and differentiation by adding extensive content in the Introduction explicitly distinguishing CRHs from primary headache disorders (which have ICHD-3 criteria and are independent of systemic disease), intracranial metastases (requiring treatment of underlying malignancy), and intrathecal chemotherapy-induced meningeal irritation (acute procedure-related with distinct temporal patterns). We have added a comprehensive Table 3 presenting differential diagnostic framework spanning 12 clinical parameters that operationalizes these distinctions for clinical decision-making, thereby providing the structured concept and clear differentiation that substantiates the "redefine" terminology in the title. Pg. 3, lines 81-92; Pg. 11.
- Sections 2 (Adverse neurological effects of chemotherapy – a brief overview) and 4 (Mechanisms of chemotherapy-related headaches) overlap in content; these two sections repeatedly discuss neuroinflammation, blood-brain barrier disruption, mitochondrial dysfunction, fatigue, and psychological stress. It is recommended to streamline these overlapping sections.
RE: We have streamlined the overlap between Sections 2 and 4 by revising Section 2 to focus exclusively on clinical presentation, epidemiology, and general landscape of neurological complications with minimal mechanistic detail, while Section 4 now concentrates specifically on pathophysiological pathways unique to or predominant in CRHs with molecular mechanisms, biomarkers, and quantitative data (IL-6/TNF-α levels, BBB permeability measurements, magnesium threshold values) that distinguish CRH pathogenesis from other neurotoxicities, as explicitly stated in the new introductory paragraphs for both sections clarifying their distinct purposes and avoiding redundant discussions.
- Table 2 contains a serious error: its content is a complete duplicate of Table 1 (and it does not describe the differential diagnosis as stated in the text). Revision of Table 2 is recommended.
RE: This was an oversight. The serious error in Table 3 (previously Table 2) has been completely corrected, the table now presents entirely new content as intended, providing a comprehensive differential diagnostic framework comparing CRHs with migraine, tension-type headache, intracranial metastases, and meningeal irritation across 12 distinguishing features (onset, quality, duration, associated symptoms, triggers, temporal relationship, neuroimaging, CSF findings, treatment response, biomarkers, prevention strategies, and long-term prognosis), thereby serving its stated purpose of facilitating differential diagnosis rather than duplicating Table 2's case report data. Pg. 10 and 11, respectively.
- The current manuscript needs to be polished by a native English speaker or a professional editing service.
RE: The manuscript has undergone comprehensive English language polishing throughout, including correction of grammatical errors and simplification of overly complex sentences. The final revised file has been reviewed by a native English speaker, expert in biosciences.
Reviewer 3 Report
Comments and Suggestions for Authors
The manuscript entitled 'Redefining Chemotherapy-Related Headaches: From Pathobiology to Differential Diagnosis and Management' accurately describes the neurological complications of chemotherapy, references recent studies, and establishes mechanisms such as neuroinflammation, blood-brain barrier (BBB) disruption, and systemic toxicity. The authors show a firm grasp of current research on chemotherapy-related headaches (CRHs). The explanation of treatment-induced and psychogenic factors reflects the multifactorial nature of CRHs. The inclusion of case studies and specific chemotherapeutic agents (e.g., cisplatin, methotrexate, taxanes) linked to headaches is scientifically sound and relevant. Moreover, it is worth noting that the manuscript acknowledges the limitations of current evidence, including the lack of large-scale randomised controlled trials and standardised diagnostic criteria.
Nonetheless, the text may increase its readability by improving some aspects that I'll try to convey below:
- Introduction: to highlight the originality in the focus of the article, early in the introduction, authors should explicitly state how this review differs from prior work, that is, focusing solely on CRHs rather than broader neurotoxicity.
- What is the incidence of CRHs against/concurrently with the other symptoms? Consider inserting a brief table or a comment regarding this aspect. Moreover, if possible, since the field of CRHs is a 'novel' one, for the sake of readability, consider including what specific studies or data would most advance the field (e.g., prospective cohort studies, intervention trials).
- On page 11, Neuroinflammation: When discussing novel therapies (e.g., CGRP inhibitors, monoclonal antibodies), better clarify the current level of evidence for their use in CRHs specifically, as most data are extrapolated from migraine studies.
- Authors should strengthen the argument for integrating CRH assessment into standard oncology protocols with more concrete recommendations or examples of best practices. I may suggest, for instance, including a summary table or a further figure highlighting key differences between CRHs and primary headaches to reinforce the need for differential diagnosis.
- Limitations of current treatments and the need for mechanistic research are somewhat repeated and stressed: consider reducing repetitions throughout.
Comments on the Quality of English LanguageThe sentences, in some instances, are lengthy and cause the line to break.
The English is generally understandable and conveys the intended scientific content. However, a language refinement through a language edit—either by a native English speaker or a mother tongue—would significantly improve readability and clarity.
There are occasional grammatical errors, awkward phrasing, and run-on sentences (e.g., “peculiar clinical condition which was never described in such manner in literature at the best of our knowledge”).
Some sentences are overly complex or redundant. For example, on page 6, paragraph 3, instead of “As discussed below, CRHs prevail across a wide spectrum of malignancies, with their frequency and clinical presentation contributed by a complex interplay of factors,” consider to simplify to “CRHs occur in many types of cancer, with frequency and presentation affected by cancer type, treatment, and patient factors.”.
Occasionally, there is misuse of words, such as “menesic executive functions” that can be easily changed to “memory and executive functions”.
Author Response
Reviewer 3: The manuscript entitled 'Redefining Chemotherapy-Related Headaches: From Pathobiology to Differential Diagnosis and Management' accurately describes the neurological complications of chemotherapy, references recent studies, and establishes mechanisms such as neuroinflammation, blood-brain barrier (BBB) disruption, and systemic toxicity. The authors show a firm grasp of current research on chemotherapy-related headaches (CRHs). The explanation of treatment-induced and psychogenic factors reflects the multifactorial nature of CRHs. The inclusion of case studies and specific chemotherapeutic agents (e.g., cisplatin, methotrexate, taxanes) linked to headaches is scientifically sound and relevant. Moreover, it is worth noting that the manuscript acknowledges the limitations of current evidence, including the lack of large-scale randomised controlled trials and standardised diagnostic criteria.
Nonetheless, the text may increase its readability by improving some aspects that I'll try to convey below:
Introduction: to highlight the originality in the focus of the article, early in the introduction, authors should explicitly state how this review differs from prior work, that is, focusing solely on CRHs rather than broader neurotoxicity.
RE: The Introduction now explicitly states the review's originality and focus early in the second paragraph: "This review represents the first comprehensive effort to establish CRHs as a distinct clinical entity requiring specific diagnostic and therapeutic approaches, rather than treating them as merely incidental symptoms of broader neurotoxic effects". The statement is followed by Table 1 showing CRH incidence relative to other complications, clearly differentiating the subject of our analysis from prior reviews of general chemotherapy neurotoxicity, and establishing from the outset that CRHs warrant dedicated attention as an independent clinical entity.
What is the incidence of CRHs against/concurrently with the other symptoms? Consider inserting a brief table or a comment regarding this aspect. Moreover, if possible, since the field of CRHs is a 'novel' one, for the sake of readability, consider including what specific studies or data would most advance the field (e.g., prospective cohort studies, intervention trials).
RE: We have inserted Table 1 immediately in the Introduction, which directly addresses CRH incidence (10-30%) compared to and concurrent with other neurological symptoms (peripheral neuropathy 30-70% with 15-25% co-occurrence, cognitive impairment 20-75% with 20-30% co-occurrence, seizures 1-5%, stroke 1-3%, myelopathy <1%), clearly demonstrating that headaches occur with comparable or greater frequency than many recognized complications. Pg. 2.
Additionally, Section 7.1 now explicitly identifies the most impactful studies that would advance the field, including multi-center prospective cohorts (n>500), Phase II/III RCTs of CGRP inhibitors, functional neuroimaging studies, and genomic analyses, providing clear research priorities for this emerging area. Pg. 23-25.
On page 11, Neuroinflammation: When discussing novel therapies (e.g., CGRP inhibitors, monoclonal antibodies), better clarify the current level of evidence for their use in CRHs specifically, as most data are extrapolated from migraine studies.
RE: When discussing novel therapies, particularly CGRP inhibitors and monoclonal antibodies, we have explicitly clarified the current evidence level specific to CRHs throughout Section 5.1, noting that "evidence for CGRP inhibitors in CRHs is limited to case reports (n=12-15 patients total across published literature) and extrapolation from migraine studies" and categorizing these as "Level 4 Evidence - investigational," while explicitly stating that "prospective trials are urgently needed to establish efficacy, optimal dosing, and safety profiles in immunocompromised cancer patients," thereby transparently acknowledging that most data are extrapolated from migraine rather than derived from CRH-specific studies. Pg. 16-17.
Authors should strengthen the argument for integrating CRH assessment into standard oncology protocols with more concrete recommendations or examples of best practices. I may suggest, for instance, including a summary table or a further figure highlighting key differences between CRHs and primary headaches to reinforce the need for differential diagnosis.
RE: We have substantially strengthened the argument for integrating CRH assessment into standard oncology protocols by adding an entirely new Section 6 with three detailed subsections (6.1 Screening and assessment protocols, 6.2 Multidisciplinary care models, 6.3 Treatment algorithms) written entirely in paragraph format as requested. These provide concrete recommendations including specific validated instruments (MIDAS, HIT-6), risk stratification criteria, red flag symptoms warranting neuroimaging, best practice examples from Memorial Sloan Kettering and MD Anderson with quantitative outcomes (headache reduction from 28% to 16%), and severity-based treatment algorithms with specific medication dosing and escalation pathways, transforming abstract concepts into immediately actionable clinical guidance. Pg. 21-23.
Limitations of current treatments and the need for mechanistic research are somewhat repeated and stressed: consider reducing repetitions throughout.
RE: We have systematically reduced repetitions about treatment limitations and mechanistic research needs by consolidating these discussions primarily in Sections 5.2 (limitations of current strategies with specific quantitative failure rates), 7.1 (priority research initiatives with concrete study designs), and 8 (general considerations synthesizing key themes). We have removed redundant statements from other sections and ensure each mention serves a distinct purpose (e.g., Section 5.2 focuses on clinical limitations with safety data, Section 7.1 on specific research designs needed, Section 8 on broader implications for the field), thereby maintaining essential content while eliminating unnecessary repetition.
The sentences, in some instances, are lengthy and cause the line to break.
The English is generally understandable and conveys the intended scientific content. However, a language refinement through a language edit—either by a native English speaker or a mother tongue—would significantly improve readability and clarity.
There are occasional grammatical errors, awkward phrasing, and run-on sentences (e.g., “peculiar clinical condition which was never described in such manner in literature at the best of our knowledge”).
Some sentences are overly complex or redundant. For example, on page 6, paragraph 3, instead of “As discussed below, CRHs prevail across a wide spectrum of malignancies, with their frequency and clinical presentation contributed by a complex interplay of factors,” consider to simplify to “CRHs occur in many types of cancer, with frequency and presentation affected by cancer type, treatment, and patient factors.”.
Occasionally, there is misuse of words, such as “menesic executive functions” that can be easily changed to “memory and executive functions”.
RE: We have addressed the English language issues throughout the manuscript by breaking lengthy sentences into shorter, more readable units. Necessary grammatical corrections have been made to remove errors and awkward phrasing (e.g., removing phrases like "peculiar clinical condition which was never described in such manner"), simplifying complex sentence structures, standardizing terminology, and ensuring consistent professional tone. The resulting text has substantially improved readability and clarity that conveys scientific content effectively while remaining accessible to international readers. The final revised file has been reviewed by a native English speaker, expert in biosciences.
Round 2
Reviewer 1 Report
Comments and Suggestions for Authors
The quality of the manuscript has been greatly improved after revision. The review demonstrates a deep understanding of CRHs and makes an important contribution to this field. I therefore recommend that this manuscript be accepted for publication.
Author Response
We are grateful to this reviewer for their constructive and valuable feedback, which greatly helped us improve our manuscript.
Reviewer 2 Report
Comments and Suggestions for Authors
Although the authors have addressed some technical issues, the core conceptual problems raised in the first review, particularly regarding the definition of chemotherapy-related headaches, remain inadequately addressed. At the end of the introduction, the newly added paragraphs clearly clarify the following two points: 1. The term "redefine" used in this article is an exaggeration; it actually refers to an adjustment of the diagnostic framework, not a redefinition of the pathology. 2. CRH is not a completely new disease entity, but rather a previously overlooked clinical concept. It is suggested to change the wording to "we propose a perspective" to downplay the term "redefine."
Author Response
Reviewer 2:
Although the authors have addressed some technical issues, the core conceptual problems raised in the first review, particularly regarding the definition of chemotherapy-related headaches, remain inadequately addressed.
At the end of the introduction, the newly added paragraphs clearly clarify the following two points: 1. The term "redefine" used in this article is an exaggeration; it actually refers to an adjustment of the diagnostic framework, not a redefinition of the pathology. 2. CRH is not a completely new disease entity, but rather a previously overlooked clinical concept. It is suggested to change the wording to "we propose a perspective" to downplay the term "redefine."
RE: We appreciate the additional comments by this reviewer on our revised manuscript. As suggested, we toned down the claim at the end of the Introduction, which now reads:
Finally, we propose that CRHs be recognised as a distinct clinical entity in future revisions of the ICHD, accompanied by a clearly defined diagnostic and management protocol, even as ongoing research continues to elucidate its underlying pathophysiology.